# Representation Deficiency in Masked Language Modeling

**Yu Meng**[1]* **Jitin Krishnan**[2] **Sinong Wang**[2] **Qifan Wang**[2] **Yuning Mao**[2]
**Han Fang**[2] **Marjan Ghazvininejad**[2] **Jiawei Han**[1] **Luke Zettlemoyer**[2]
[1]University of Illinois Urbana-Champaign [2]Meta AI
[1]{yumeng5, hanj}@illinois.edu [2]{jitinkrishnan, sinongwang,
 wqfcr, yuningm, hanfang, ghazvini, lsz}@meta.com

## Abstract

Masked Language Modeling (MLM) has been one of the most prominent approaches for pretraining bidirectional text encoders due to its simplicity and effectiveness. One notable concern about MLM is that the special [MASK] symbol causes a discrepancy between pretraining data and downstream data as it is present only in pretraining but not in fine-tuning. In this work, we offer a new perspective on the consequence of such a discrepancy: We demonstrate empirically and theoretically that MLM pretraining allocates some model dimensions exclusively for representing [MASK] tokens, resulting in a representation deficiency for real tokens and limiting the pretrained model's expressiveness when it is adapted to downstream data without [MASK] tokens. Motivated by the identified issue, we propose MAE-LM, which pretrains the Masked Autoencoder architecture with MLM where [MASK] tokens are excluded from the encoder. Empirically, we show that MAE-LM improves the utilization of model dimensions for real token representations, and MAE-LM consistently outperforms MLM-pretrained models on the GLUE and SQuAD benchmarks.

## 1 Introduction

Pretraining text encoders to learn from bidirectional contexts has achieved enormous success in various natural language processing (NLP) tasks (Clark et al., 2020; Devlin et al., 2019; Liu et al., 2019). Masked Language Modeling (MLM) (Devlin et al., 2019) is among one of the most prominent pretraining approaches due to its conceptual simplicity and empirical effectiveness: By randomly masking a portion of input tokens and training a Transformer encoder to predict the original content based on the remaining bidirectional contexts, the model learns robust representations that generalize well to diverse downstream tasks. Besides its broad impact in NLP, MLM has also been widely adopted for pretraining in other domains, such as images (Bao et al., 2022; Xie et al., 2022), videos (Tong et al., 2022; Wang et al., 2022) and graphs (Hou et al., 2022).

Despite its remarkable success, the effectiveness of MLM may be hindered by a discrepancy between pretraining and fine-tuning: The special [MASK] token occurs only in pretraining but not in downstream tasks. While a few previous studies (Clark et al., 2020; Yang et al., 2019) have attempted to address this issue, they end up proposing new training objectives instead of systematically investigating why and how such a discrepancy impacts the generalization of MLM-pretrained models.

In this work, we study the consequence of including [MASK] tokens in MLM pretraining by examining the learned token representation space. We empirically and theoretically show that [MASK] token representations exclusively occupy some model dimensions, thereby reducing the model capacity for representing real tokens. Such a representation deficiency issue may not be simply addressed by fine-tuning on downstream tasks: Those dimensions exclusively used for [MASK] tokens have not been pretrained to represent real tokens, and will have to be either trained from scratch on downstream data, raising the risk of overfitting (Hendrycks et al., 2019; Kumar et al., 2022), or become unused, resulting in a waste of model capacity.

---

*Work done during internship at Meta AI.

To address the representation deficiency issue, we propose a simple text encoder pretraining method, MAE-LM, which conducts MLM pretraining based on the Masked Autoencoder architecture (He et al., 2022). Notably, `[MASK]` tokens are omitted from the encoder's input so that the real token representations can utilize the entire model dimensions theoretically. An auxiliary decoder, used only in pretraining and not in fine-tuning, takes the encoder's output representations and `[MASK]` positions to predict the original tokens. We demonstrate empirically that by excluding `[MASK]` tokens from the encoder, MAE-LM improves the utilization of model dimensions both in pretraining and downstream tasks and achieves consistent and notable improvements over previous models pretrained by MLM and its variants on the GLUE and SQuAD benchmarks.[1]

Our main contributions are as follows: (1) We investigate the token representation space trained by MLM, and identify a previously unknown representation deficiency issue when the pretrained model is applied to real data without `[MASK]` tokens. (2) Based on empirical and theoretical analyses, we explain why the representation deficiency issue occurs in the conventional MLM pretraining setup. (3) We show that a simple pretraining method MAE-LM can address the identified issue and improve the downstream task performance of previous MLM-pretrained models under multiple pretraining and fine-tuning settings.

## 2 Analysis of Token Representations in MLM

### 2.1 Preliminaries

**Transformer Encoder.** Transformer encoders contain multiple Transformer layers, where each layer consists of two submodules, multi-head self-attention (MHSA) and feed-forward network (FFN). The self-attention mechanism uses queries $Q$ and keys $K$ to compute attention weights, and outputs a weighted sum of the values $V$. MHSA performs self-attention in parallel over $N$ heads as follows:

$$\text{Attn}(Q, K, V) = \text{Softmax}\left(\frac{QK^\top}{\sqrt{d_h}}\right) V,$$

$$\text{MHSA}(X) = \text{Concat}(\text{head}_1, \ldots, \text{head}_N)W^O, \quad \text{head}_h = \text{Attn}(XW_h^Q, XW_h^K, XW_h^V),$$

where $X \in \mathbb{R}^{n \times d}$ is the input representations to MHSA, $n$ is the number of tokens and $d$ is the model dimension. $d_h$ is the dimension of head $h$ and is usually set to $d/N$. $W_h^Q, W_h^K, W_h^V \in \mathbb{R}^{d \times d_h}$ and $W^O \in \mathbb{R}^{d \times d}$ are learnable weight matrices. The outputs of MHSA are further passed to FFN which learns nonlinear transformations to derive the final outputs of the Transformer layer.

**Masked Language Modeling (MLM).** Given a text sequence $x = [x_1, \ldots, x_i, \ldots, x_n]$, MLM randomly replaces a set of token positions $\mathcal{M}$ with `[MASK]` symbols. The resulting partially masked sequence $\hat{x} = [x_1, \ldots, \text{[MASK]}_i, \ldots, x_n]$ is then fed to the Transformer encoder $\theta$ which outputs the token representations $H = [h_1, \ldots, h_i, \ldots, h_n]$. The encoder $\theta$ is trained to predict the original token out of the vocabulary $\mathcal{V}$ at each masked position by minimizing the cross-entropy loss $\mathcal{L}_{\text{MLM}}$:

$$p_\theta(x_i|\hat{x}) = \frac{\exp(e_{x_i}^\top h_i)}{\sum_{x' \in \mathcal{V}} \exp(e_{x'}^\top h_i)}, \quad \mathcal{L}_{\text{MLM}} = \mathbb{E}\left(-\sum_{i \in \mathcal{M}} \log p_\theta\left(x_i|\hat{x}\right)\right), \tag{1}$$

where $e_x$ refers to the embedding of token $x$.

### 2.2 Rank-Deficient Real Token Representations

MLM pretraining introduces a special `[MASK]` token to replace the token positions to be predicted, but such `[MASK]` tokens are usually absent from downstream task data. Therefore, to study the PLM's capacity for downstream data representation, we examine the *real token* representation space trained with MLM. A common measure of the representation space capacity is the *rank* of the data representation matrix (Ansuini et al., 2019; Bhojanapalli et al., 2020). In our case, this refers to the real token representation matrix $H_\mathcal{R} \in \mathbb{R}^{n \times d}$ ($n \gg d$) where each row corresponds to the representation of a real token. Ideally, one would hope $H_\mathcal{R}$ to have high column rank (*i.e.*, $\text{rank}(H_\mathcal{R}) \approx d$) so that more model dimensions are effective for modeling real tokens. However, as we will show next, a

---

[1]Code can be found at `https://github.com/yumeng5/MAE-LM`.

portion of the model dimensions will be exclusively used for `[MASK]` token representations in MLM pretraining, so that $\boldsymbol{H}_{\mathcal{R}}$ is necessarily rank-deficient (*i.e.*, not all model dimensions are leveraged to represent real tokens).

**Empirical Evidence.** We evaluate the representation space of a pretrained 12-layer RoBERTa$_{\text{base}}$ model (Liu et al., 2019) on the validation set of the pre-training corpus with 5 million tokens. We first apply $15\%$ random masks to these input sequences (same as the pretraining setting), and obtain the token representation matrix $\boldsymbol{H}^l \in \mathbb{R}^{n \times d}$ ($n \approx 5 \times 10^6$ is the total number of tokens in the corpus, $d = 768$ is the model dimension), which contains both real token and mask token representations, for each layer $l$ in the pretrained RoBERTa. We then feed the same input sequences in their original form (*i.e.*, without `[MASK]`) to the pretrained RoBERTa model and obtain the token representation matrix $\widetilde{\boldsymbol{H}}^l \in \mathbb{R}^{n \times d}$ which consists of real token representations only. Comparing the rank of $\widetilde{\boldsymbol{H}}^l$ with $\boldsymbol{H}^l$ gives insights about the change in representation capacity when adapting a pretrained MLM model to inputs without `[MASK]`.

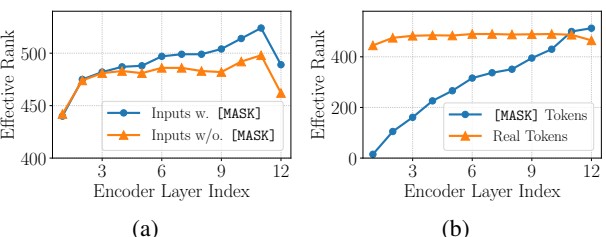

Figure 1: In an MLM-pretrained model, (a) some model dimensions are exclusively used for representing `[MASK]` tokens, resulting in a representation deficiency for modeling inputs without `[MASK]`, especially in deeper layers; (b) the effective rank of `[MASK]` token representation space increases throughout Transformer layers.

Since numerical errors and small perturbations practically render any large matrix full-rank regardless of its actual rank, we compute the *effective rank* (Cai et al., 2021) of a matrix $\boldsymbol{H}$: We only consider $\boldsymbol{H}$'s most significant components that account for the majority of the variance reflected by singular values. Given a threshold value $\tau$, we define the $\tau$-effective rank of $\boldsymbol{H}$ as $\text{rank}_\tau(\boldsymbol{H}) = \arg\min_k \left( \frac{\sum_{i=1}^k \sigma_i^2}{\sum_{i=1}^d \sigma_i^2} \geq \tau \right)$, where $\sigma_i$ is the $i$th largest singular value of $\boldsymbol{H}$. For example, $\text{rank}_{0.9}(\boldsymbol{H}) = 10$ means that $90\%$ of $\boldsymbol{H}$'s variance can be captured with 10 dimensions. We follow the definition of effective rank in Cai et al. (2021) only to perform empirical computations of the rank to showcase the issue, and we do not use it in our theoretical analysis below.

Figure 1(a) shows $\text{rank}_{0.9}(\boldsymbol{H}^l)$ (Input w. `[MASK]`) and $\text{rank}_{0.9}(\widetilde{\boldsymbol{H}}^l)$ (Input w/o. `[MASK]`). It generally holds that $\text{rank}_{0.9}(\widetilde{\boldsymbol{H}}^l) < \text{rank}_{0.9}(\boldsymbol{H}^l)$, and the gap is more prominent in deeper layers. This demonstrates that some model dimensions are reserved for `[MASK]` token representations in almost all encoder layers, and these dimensions are not active when the input sequences consist of real tokens entirely. Such representation deficiencies for modeling real tokens become more severe in deeper layers where `[MASK]` token representations occupy more dimensions, shown in Figure 1(b).

**Theoretical Analysis.** We theoretically validate the empirical observation above that MLM necessarily allocates a subspace for `[MASK]` token representations which is not contained by the real token representation subspace, so that the real token representations are rank-deficient.

**Lemma 2.1** (Rank increase of `[MASK]` token representations in Transformer encoder). *The rank of `[MASK]` token representations will increase from the input layer to the output layer of an L-layer Transformer encoder trained with MLM (i.e., $rank(\boldsymbol{H}_{\mathcal{M}}^L) \gg rank(\boldsymbol{H}_{\mathcal{M}}^0)$).*

*Proof.* We first show that $\boldsymbol{H}_{\mathcal{M}}^L$ will be high-rank in a well-trained MLM model and then show that $\boldsymbol{H}_{\mathcal{M}}^0$ is necessarily low-rank, and thus the statement holds.

As shown in Equation (1), the output token probability distributions at masked positions are computed from the encoder's output representations $\boldsymbol{H}_{\mathcal{M}}^L \in \mathbb{R}^{m \times d}$ and token embeddings $\boldsymbol{E} \in \mathbb{R}^{|\mathcal{V}| \times d}$. Denote the true log probability distributions of the masked token prediction task as $\boldsymbol{T} \in \mathbb{R}^{m \times |\mathcal{V}|}$:

$$
\boldsymbol{T} = \begin{bmatrix}
\log p\left(x_1|\hat{\boldsymbol{x}}_1\right) & \log p\left(x_2|\hat{\boldsymbol{x}}_1\right) & \cdots & \log p\left(x_{|\mathcal{V}|}|\hat{\boldsymbol{x}}_1\right) \\
\log p\left(x_1|\hat{\boldsymbol{x}}_2\right) & \log p\left(x_2|\hat{\boldsymbol{x}}_2\right) & \cdots & \log p\left(x_{|\mathcal{V}|}|\hat{\boldsymbol{x}}_2\right) \\
\vdots & \vdots & \ddots & \vdots \\
\log p\left(x_1|\hat{\boldsymbol{x}}_m\right) & \log p\left(x_2|\hat{\boldsymbol{x}}_m\right) & \cdots & \log p\left(x_{|\mathcal{V}|}|\hat{\boldsymbol{x}}_m\right)
\end{bmatrix},
$$

then $\boldsymbol{H}_{\mathcal{M}}^{L}$ and $\boldsymbol{E}$ are trained to approximate $\boldsymbol{T}$ with a row shift (due to the softmax normalization) (Yang et al., 2018):

$$\boldsymbol{H}_{\mathcal{M}}^{L}\boldsymbol{E}^{\top} \approx \boldsymbol{T} + \boldsymbol{c}\mathbf{1}^{\top}, \tag{2}$$

where $\boldsymbol{c} \in \mathbb{R}^{m}$ contains the shifting constant added to each row, and $\mathbf{1} \in \mathbb{R}^{|\mathcal{V}|}$ is a vector of all ones.

It is shown in Yang et al. (2018) that the true probability distribution $\boldsymbol{T}$ is high-rank (as high as $|\mathcal{V}|$) due to the complexity of natural language. Since $\text{rank}(\boldsymbol{H}_{\mathcal{M}}^{L}\boldsymbol{E}^{\top}) \leq \min\{\text{rank}(\boldsymbol{H}_{\mathcal{M}}^{L}), \text{rank}(\boldsymbol{E})\}$, both $\boldsymbol{H}_{\mathcal{M}}^{L}$ and $\boldsymbol{E}$ need to be high-rank to achieve a good approximation of $\boldsymbol{T} + \boldsymbol{c}\mathbf{1}^{\top}$.

Next, we show $\boldsymbol{H}_{\mathcal{M}}^{0}$ is low-rank. $\boldsymbol{H}_{\mathcal{M}}^{0}$ is the sum of token embeddings and position embeddings at masked positions:

$$\boldsymbol{H}_{\mathcal{M}}^{0} = \mathbf{1}\boldsymbol{e}_{[\texttt{MASK}]}^{\top} + \boldsymbol{P},$$

where $\boldsymbol{e}_{[\texttt{MASK}]} \in \mathbb{R}^{d}$ is the [MASK] token embedding, and $\boldsymbol{P} \in \mathbb{R}^{m \times d}$ is the position embeddings.

Since we have $\text{rank}(\mathbf{1}\boldsymbol{e}_{[\texttt{MASK}]}^{\top} + \boldsymbol{P}) \leq \text{rank}(\mathbf{1}\boldsymbol{e}_{[\texttt{MASK}]}^{\top}) + \text{rank}(\boldsymbol{P}) = \text{rank}(\boldsymbol{P}) + 1$, we only need to show $\boldsymbol{P}$ is low-rank. Previous studies (He et al., 2021; Ke et al., 2021) have identified that position embeddings $\boldsymbol{P}$ and token embeddings $\boldsymbol{E}$ encode disjoint information, and are learned in separate subspaces of $\mathbb{R}^{d}$. Therefore, $\text{rank}(\boldsymbol{P}) \leq d - \text{rank}(\boldsymbol{E})$. We also showed that $\boldsymbol{E}$ must be high-rank to satisfy Equation (2), and thus $\boldsymbol{P}$ is necessarily low-rank. Finally, $\boldsymbol{H}_{\mathcal{M}}^{0}$ is also low-rank as $\text{rank}(\boldsymbol{H}_{\mathcal{M}}^{0}) \leq \text{rank}(\boldsymbol{P}) + 1$. □

*Remark.* Lemma 2.1 corresponds to the empirical observation in Figure 1(b), and can be intuitively interpreted as a necessary consequence of the [MASK] token contextualization process in Transformers: The [MASK] representations at the input layer are context-free, and they need to aggregate contextual information from other tokens in the sequence for predicting the original word, resulting in an increase in the information content of [MASK] token representations. We also note that the rank increase statement does not necessarily apply to real token representations. This is because MLM does not directly train the real token representations (*e.g.*, the training objective in Equation (2) does not apply to real token positions[2]).

Based on Lemma 2.1, we proceed to prove that $\boldsymbol{H}_{\mathcal{M}}^{l}$ occupies a different subspace that is not contained by the subspace of $\boldsymbol{H}_{\mathcal{R}}^{l}$, resulting in deficient representations for real tokens. In the following, we analyze the rank change induced by the *self-attention* mechanism since it is the source of contextualization of [MASK] tokens, and the effectiveness of text encoders is typically attributed to the contextualized representations (Ethayarajh, 2019). While we do not account for MLPs and residual connections, our analysis validates that the rank deficiency is caused by the self-attention mechanism, and in practice, MLPs and residual connections do not prevent the issue from happening.

**Theorem 2.2** (Rank deficiency of real token representations). *There exists some layer $l$ in the Transformer encoder where the real token representation $\boldsymbol{H}_{\mathcal{R}}^{l}$ is rank-deficient. In particular, the row space of $\boldsymbol{H}_{\mathcal{R}}^{l}$ does not contain the row space of [MASK] token representation $\boldsymbol{H}_{\mathcal{M}}^{l}$.*

*Proof.* We provide a proof sketch below. Detailed proofs can be found in Appendix A. We prove the statement by contradiction: Suppose that the row space of $\boldsymbol{H}_{\mathcal{R}}^{l} \in \mathbb{R}^{n \times d}$ contains the row space of $\boldsymbol{H}_{\mathcal{M}}^{l} \in \mathbb{R}^{m \times d}$, then we can represent $\boldsymbol{H}_{\mathcal{M}}^{l}$ with $\boldsymbol{H}_{\mathcal{R}}^{l}$ via a linear combination weight matrix $\boldsymbol{U}$:

$$\boldsymbol{H}_{\mathcal{M}}^{l} = \boldsymbol{U}\boldsymbol{H}_{\mathcal{R}}^{l}, \quad \boldsymbol{U} \in \mathbb{R}^{m \times n}. \tag{3}$$

We show that under this assumption, $\boldsymbol{H}_{\mathcal{M}}^{l}$ will converge exponentially (with $l$) to a rank-1 matrix, which contradicts with Lemma 2.1. To examine the matrix rank, we follow the definition of matrix residual $\boldsymbol{R}^{l}$ (Dong et al., 2021) which measures the difference between $\boldsymbol{H}_{\mathcal{R}}^{l}$ and a rank-1 matrix:

$$\boldsymbol{R}^{l} = \boldsymbol{H}_{\mathcal{R}}^{l} - \mathbf{1}\boldsymbol{h}^{\top}, \quad \boldsymbol{h} = \arg\min_{\boldsymbol{x}} \left\| \boldsymbol{H}_{\mathcal{R}}^{l} - \mathbf{1}\boldsymbol{x}^{\top} \right\|.$$

---

[2]Some MLM training settings adopt a trick that keeps 10% of [MASK] as original tokens and randomly replaces another 10% of [MASK] with other tokens. Even with this trick, the training signals on real token representations are scarce. Furthermore, later studies (Wettig et al., 2023) report that this trick is not necessary—training exclusively on [MASK] positions performs well.

Based on the self-attention formula and the assumption in Equation (3), we can derive a bound for the norm of $\boldsymbol{R}^l$ as a function of $\boldsymbol{R}^{l-1}$:

$$\left\|\boldsymbol{R}^l\right\|_{1,\infty} \leq 4\epsilon \left\|\boldsymbol{R}^{l-1}\right\|_{1,\infty}^3, \quad \epsilon = \left\|\frac{\boldsymbol{W}^Q \boldsymbol{W}^{K^\top}}{\sqrt{d}}\right\|_1 \left\|\boldsymbol{W}^V \boldsymbol{W}^O\right\|_{1,\infty} \|\boldsymbol{U}\|_\infty \left(1 + \|\boldsymbol{U}\|_\infty\right).$$

where $\|\cdot\|_{1,\infty}$ denotes the geometric mean of $\ell_1$ and $\ell_\infty$ norm. This shows that $\left\|\boldsymbol{R}^l\right\|_{1,\infty}$ converges exponentially with $l$ to zero, and thus $\boldsymbol{H}_\mathcal{R}^l$ converges exponentially with $l$ to a rank-1 matrix. We also have $\text{rank}(\boldsymbol{H}_\mathcal{M}^l) \leq \text{rank}(\boldsymbol{H}_\mathcal{R}^l)$ as the row space of $\boldsymbol{H}_\mathcal{M}^l$ is contained by the row space of $\boldsymbol{H}_\mathcal{R}^l$. Hence, $\boldsymbol{H}_\mathcal{M}^l$ will also converge exponentially to a rank-1 matrix, which contradicts with Lemma 2.1. Therefore, the statement holds. $\square$

*Remark.* Theorem 2.2 demonstrates that at least some [MASK] token representations and real token representations need to be linearly independent so that the rank of $\boldsymbol{H}_\mathcal{M}^l$ may increase through encoder layers. As a result, the real token representation $\boldsymbol{H}_\mathcal{R}^l$ cannot utilize the entire model dimensions and is prone to rank deficiency.

## 3    MAE-LM: MASKED AUTOENCODERS FOR MLM

To address the representation deficiency issue in MLM, we propose a simple framework MAE-LM, which pretrains bidirectional Transformer encoders using the MLM objective, but based on the Masked Autoencoder (He et al., 2022; Liao et al., 2022) structure.   An overview of MAE-LM is shown in Figure 2. While previous applications of the architecture are mainly motivated by the efficiency benefit of reduced input sequence lengths, its effects on the learned tokens representations have not been thoroughly studied.

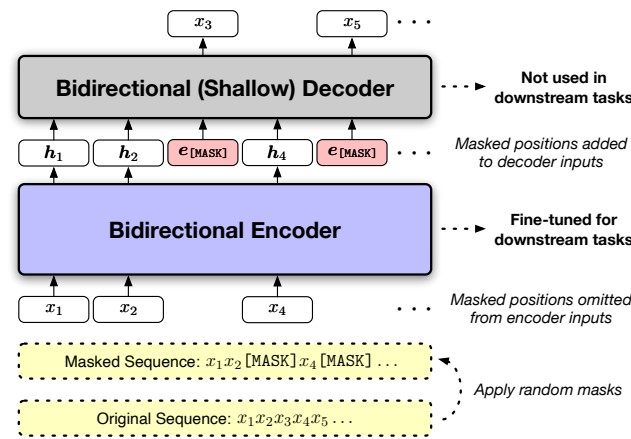

Figure 2: Overview of MAE-LM. Masked positions are omitted from encoder inputs so that the encoder purely models real tokens. A shallow decoder takes the encoder's output representations and masked positions to predict the original tokens. After pretraining, only the encoder (but not the decoder) is fine-tuned for downstream tasks.

**Excluding** [MASK] **from the Encoder.** An important design in MAE-LM is that [MASK] tokens are excluded from the encoder inputs so that no model dimensions will be used to represent [MASK] tokens. Hence, the representations of real tokens $\boldsymbol{H}_\mathcal{R}$ can theoretically utilize the entire space $\mathbb{R}^d$, which addresses the representation bottleneck in conventional MLM pretraining. Specifically, given a masked sequence $\hat{\boldsymbol{x}} = [x_1, \ldots, \text{[MASK]}_i, \ldots, x_n]$ and let $\mathcal{M}$ denote the set of masked positions, the encoder's input sequence $\boldsymbol{H}^0$ consists of the sum of token embeddings $\boldsymbol{e}_{x_i}$ and position embeddings $\boldsymbol{p}_i$ at real token positions $i \notin \mathcal{M}$:

$$\boldsymbol{H}^0 = \left\{\boldsymbol{h}_i^0\right\}_{i \notin \mathcal{M}}, \quad \boldsymbol{h}_i^0 = \boldsymbol{e}_{x_i} + \boldsymbol{p}_i.$$

**Decoder Configuration.** In order to predict the original tokens at masked positions, the encoder's output token representations $\boldsymbol{H}^L$ are further passed to an auxiliary bidirectional decoder. While standard Transformer decoders perform unidirectional self-attention (and cross-attention to encoder outputs) for autoregressive decoding, our decoder performs bidirectional self-attention (same as the encoder). It is called a decoder as it takes encoded representations as input and outputs tokens. The decoder's input sequence $\widehat{\boldsymbol{H}}^0$ needs to include the [MASK] token embedding $\boldsymbol{e}_{\text{[MASK]}}$ and position embeddings $\boldsymbol{p}_i$ so that the decoder is aware of the positions to be predicted:

$$\widehat{\boldsymbol{H}}^0 = \left\{\widehat{\boldsymbol{h}}_i^0\right\}_{1 \leq i \leq n}, \quad \widehat{\boldsymbol{h}}_i^0 = \begin{cases} \boldsymbol{e}_{\text{[MASK]}} + \boldsymbol{p}_i & i \in \mathcal{M} \\ \boldsymbol{h}_i^L + \boldsymbol{p}_i & i \notin \mathcal{M} \end{cases}.$$

Table 1: Standard single-task, single-model fine-tuning results (medians over five random seeds) evaluated on GLUE and SQuAD 2.0 development sets. Results not available in prior research are marked with "–". We use Spearman correlation for STS, Matthews correlation for CoLA, and accuracy for the other tasks on GLUE. The "AVG" column contains the averaged results across the eight GLUE tasks. All baseline results are taken from public reports unless marked with (Ours).

| Model | GLUE (Single-Task) | | | | | | | | | SQuAD 2.0 | |
| | MNLI-(m/mm) | QQP | QNLI | SST-2 | CoLA | RTE | MRPC | STS-B | AVG | EM | F1 |
|---|---|---|---|---|---|---|---|---|---|---|---|
| *base* setting: Pretrained on Wikipedia & Book Corpus (16GB) | | | | | | | | | | | |
| BERT | 84.5/- | 91.3 | 91.7 | 93.2 | 58.9 | 68.6 | 87.3 | 89.5 | 83.1 | 73.7 | 76.3 |
| ALBERT | 81.6/- | – | – | 90.3 | – | – | – | – | – | 77.1 | 80.0 |
| UniLMv2 | 86.1/86.1 | – | – | 93.2 | – | – | – | – | – | 80.9 | 83.6 |
| TUPE | 86.2/86.2 | 91.3 | 92.2 | 93.3 | 63.6 | 73.6 | 89.9 | 89.2 | 84.9 | – | – |
| RoBERTa | 84.7/- | – | – | 92.7 | – | – | – | – | – | – | 79.7 |
| RoBERTa (Ours) | 85.9/85.8 | **91.6** | 92.3 | 93.7 | **64.3** | 75.5 | 88.7 | 89.5 | 85.2 | 78.3 | 81.5 |
| MAE-LM | **87.2/87.1** | **91.6** | **92.9** | **93.8** | 63.1 | **79.1** | **90.2** | **90.9** | **86.1** | **81.1** | **84.1** |
| *base++* setting: Pretrained on larger pretraining corpora (160GB) | | | | | | | | | | | |
| ALBERT | 82.4/- | – | – | 92.8 | – | – | – | – | – | 76.3 | 79.1 |
| RoBERTa | 87.6/- | **91.9** | 92.8 | 94.8 | 63.6 | 78.7 | 90.2 | 91.2 | 86.4 | 80.5 | 83.7 |
| UniLMv2 | 88.5/- | 91.7 | 93.5 | **95.1** | 65.2 | 81.3 | **91.8** | 91.0 | 87.1 | 83.3 | 86.1 |
| MAE-LM | **89.1/89.1** | 91.7 | **93.8** | **95.1** | **65.9** | **85.2** | 90.2 | **91.6** | **87.8** | **83.5** | **86.5** |

The decoder's output representations will be trained with the MLM objective shown in Equation (1). Since the decoder includes `[MASK]` tokens, it is subject to the representation deficiency for modeling real tokens as analyzed in Section 2. Therefore, the decoder is *not* used in fine-tuning on downstream tasks. The decoder is made to be shallow (the decoder depth is $1/6 - 1/3$ of the encoder in our experiments) not only for pretraining efficiency, but also to push the encoder to learn robust token representations—if the decoder is too strong, it alone may learn the MLM task well without requiring good encoder representations $\boldsymbol{H}^L$.

Despite using an additional decoder in pretraining, MAE-LM's pretraining time cost is roughly equal to that of conventional MLM pretraining (*e.g.*, RoBERTa). This is because the exclusion of `[MASK]` tokens from the encoder practically reduces its input sequence length (*e.g.*, $15\%$ random masks shorten the encoder's input length by $15\%$), bringing down the encoder's computation cost.

## 4 EXPERIMENTS

### 4.1 PRETRAINING AND EVALUATION SETUP

**Pretraining Settings.** We evaluate MAE-LM mainly under the base model scale for two pretraining settings: *base* and *base++*. Both settings pretrain 12-layer Transformers with 768 model dimensions. The *base* setting uses 16GB training corpus following BERT (Devlin et al., 2019) while the *base++* setting uses 160GB training corpus following RoBERTa (Liu et al., 2019). The details can be found in Appendix D. Additional results of larger model scales are presented in Appendix E. All settings use the MLM objective for pretraining without any sequence-level tasks.

**Downstream Tasks and Fine-Tuning.** We evaluate the pretrained models on the GLUE (Wang et al., 2018) and SQuAD 2.0 (Rajpurkar et al., 2018) benchmarks. The details about GLUE tasks can be found in Appendix B. We adopt standard fine-tuning as in BERT (Devlin et al., 2019) and RoBERTa (Liu et al., 2019). The hyperparameter search space for fine-tuning can be found in Appendix D. All reported fine-tuning results are the medians of five random seeds on GLUE and SQuAD, following previous studies (Liu et al., 2019). Additional few-shot and zero-shot evaluation results are presented in Appendix E.

**Baselines.** We compare with various baselines pretrained by MLM (and variants of MLM) under each setting, including BERT (Devlin et al., 2019), ALBERT (Lan et al., 2020), UniLMv2 (Bao et al., 2020), TUPE (Ke et al., 2021), and RoBERTa (Liu et al., 2019). The baseline results, unless marked by "(Ours)", are taken from the original papers. To eliminate the performance difference due to implementation details and computation environment, we also pretrain and fine-tune RoBERTa (the most important baseline) under exactly the same *base* pretraining setting with MAE-LM, which is denoted with "RoBERTa (Ours)".

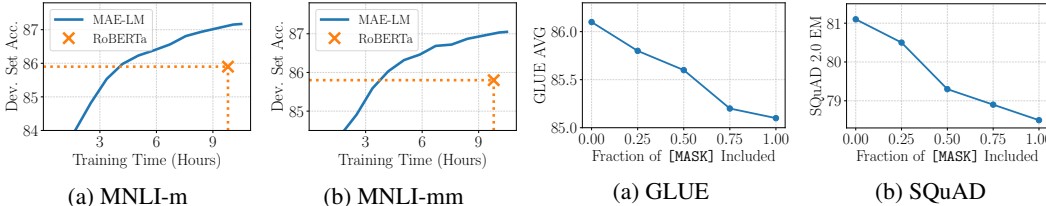

Figure 3: MNLI dev set accuracy by fine-tuning intermediate MAE-LM$_{base}$ checkpoints at different time steps. We also mark the pretraining time and final performance of RoBERTa (Ours).

Figure 4: GLUE average scores and SQuAD EM scores when different fractions of [MASK] tokens are included in the input sequences to the encoder of MAE-LM$_{base}$.

## 4.2 OVERALL RESULTS

Table 1 shows the results under the two base model pretraining settings on the GLUE and SQuAD 2.0 benchmarks. Overall, MAE-LM outperforms previous models pretrained by MLM and its variants. Notably, the gains of MAE-LM over RoBERTa (the standard MLM pretrained model) are quite consistent across tasks and pretraining settings.

**Pretraining Efficiency.** In Figure 3, we illustrate MAE-LM$_{base}$'s fine-tuning performance when pretrained for different amounts of time. MAE-LM takes slightly more time than RoBERTa when trained on the same amount of data, but to reach RoBERTa's MNLI accuracy, MAE-LM only needs about 40% of its pretraining time.

## 4.3 ABLATION STUDIES

Table 2 shows several groups of ablations to study the important components in MAE-LM.

**Naive Baselines.** To validate that the effectiveness of MAE-LM is not from simply using the additional decoder in pretraining, we first compare two naive baselines: (1) the standard MLM (enc. w. [MASK]) and (2) adding the same decoder used in MAE-LM but still pretrains the encoder with [MASK] tokens included in inputs (enc. w. [MASK] + dec.). The two baselines perform similarly, confirming that naively using the decoder does not benefit downstream tasks.

**Handling [MASK].** We compare with other ways of handling [MASK] tokens in the encoder: (1) including [MASK] in encoder's inputs but resetting [MASK] token positions to the [MASK] token embedding $e_{[MASK]}$ in decoder's inputs (enc. w. [MASK], dec. resets [MASK]) and (2) randomly replacing [MASK] tokens in encoder's inputs with other real tokens from the vocabulary (random replace w. real token). The first variation improves the performance over vanilla MLM, showing that when [MASK] is present in the encoder, resetting the [MASK] token embeddings in the decoder helps. This validates our analysis in Theorem 2.2 that the rank increase of [MASK] token representations is the main cause of representation deficiency, and preventing [MASK] token representations in the encoder from being explicitly trained is one way to mitigate the issue, though it is slightly worse than completely excluding [MASK] from the encoder. The second variation demonstrates that replacing [MASK] tokens with random real tokens, though avoiding the

Table 2: Ablations evaluated with GLUE average scores. The setting of MAE-LM$_{base}$ is: enc. w/o. [MASK]; aligned position encoding w. relative position encoding; bi. self-attention; 4 layer, 768 dimension.

| Group | Setting | GLUE |
|---|---|---|
| **Original** | MAE-LM$_{base}$ | 86.1 |
| **Naive** | enc. w. [MASK] (*i.e.*, MLM) | 85.2 |
| | enc. w. [MASK] + dec. | 85.1 |
| **Handling** [MASK] | enc. w. [MASK], dec. resets [MASK] | 85.9 |
| | random replace w. real token | 85.1 |
| **Position Encoding** | misaligned position encoding | 86.0 |
| | no relative position encoding | 86.1 |
| **Decoder Attention** | bi. self-attention + cross-attention | 85.4 |
| | uni. self-attention + cross-attention | 85.5 |
| | cross-attention | 86.0 |
| **Decoder Size** | 2 layer, 768 dimension | 85.8 |
| | 6 layer, 768 dimension | 84.8 |
| | 4 layer, 512 dimension | 85.8 |
| | 4 layer, 1024 dimension | 85.5 |

representation deficiency problem, worsens the context quality in pretraining. On balance, it does not yield better results than MLM.

**Position Encoding.** MAE-LM aligns the position encoding based on each token's position in the original sequence, and the position indices of masked positions are skipped. MAE-LM also uses relative position encoding (Raffel et al., 2019). We create two ablations: (1) apply consecutive position encoding that does not reflect the masked positions (misaligned position encoding); and (2) remove the relative position encoding from MAE-LM (no relative position encoding). Overall, the variations in position encoding do not result in notable performance differences.

**Decoder Attention.** MAE-LM uses bidirectional self-attention in the decoder. We compare with other decoder attention configurations: (1) additionally use cross-attention to encoder's output representations (bi. self-attention + cross-attention); (2) use unidirectional self-attention and cross-attention for autoregressive decoding of the entire sequence, similar to BART (Lewis et al., 2020a) (uni. self-attention + cross-attention); and (3) only use cross-attention (cross-attention). Bidirectional self-attention only in the decoder is simple and performs the best.

**Decoder Size.** MAE-LM uses a 4-layer decoder with the same dimensionality (768) as the encoder. We experiment with other decoder sizes (when the decoder's dimension is different from the encoder, we add a linear projection between the encoder's output and the decoder's input): (1) 2-layer, 768 dimension; (2) 6-layer, 768 dimension; (3) 4-layer, 512 dimension; and (4) 4-layer, 1024 dimension. Overall, using a relatively small decoder yields good results.

**Gradual Transition from MAE-LM to Standard MLM.** To further examine the empirical benefits of excluding [MASK] tokens from MAE-LM's encoder, we create a set of "stepping stones" between MAE-LM and standard MLM as follows: Out of all [MASK] tokens in the sequence $\hat{x}$, we include a fraction ($\delta$) of them in the encoder's input sequence. The rest $(1 - \delta)$ of [MASK] tokens are excluded from the encoder's input and added to the decoder's input. Then $\delta = 0$ represents MAE-LM, and $\delta = 1$ refers to the standard MLM[3]. Figure 4 illustrates the fine-tuning performance changes on GLUE and SQuAD as we transition from MAE-LM to standard MLM. There is a clear trend that including a higher portion of [MASK] tokens in the encoder degrades its performance.

### 4.4 MAE-LM Improves Model Dimension Utilization

To further validate the effectiveness of MAE-LM in improving the utilization of model dimensions for representing real tokens, we compute the 0.9-effective rank of the encoder's token representations $\text{rank}_{0.9}(\boldsymbol{H}^L)$ both after pretraining (evaluated on the validation set of the pretraining corpus) and after further fine-tuning on MNLI. Figure 5(a) shows the effective rank throughout encoder layers for (1) RoBERTa when the inputs contain [MASK] (MLM w. [MASK]); (2) RoBERTa when the inputs are all real tokens (MLM w/o. [MASK]); and (3) MAE-LM. MAE-LM closes the gap

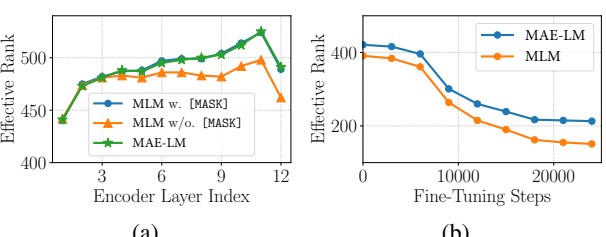

(a)                              (b)

Figure 5: (a) MAE-LM effectively closes the rank gap in vanilla MLM with inputs containing or not containing [MASK]. (b) During fine-tuning, the advantage in effective rank of MAE-LM over vanilla MLM still holds.

caused by [MASK] tokens in vanilla MLM pretraining. Figure 5(b) further validates that MAE-LM maintains its advantage in the effective rank of real token representations during fine-tuning on MNLI. This highlights the importance of addressing the representation deficiency issue in pretraining: The model dimensions not pretrained to represent real tokens may not be easily leveraged in fine-tuning.

## 5 Related Work

**Language Model Pretraining.** Various pretraining methods have been proposed for different purposes: Standard autoregressive language modeling (Brown et al., 2020; Radford et al., 2018;

---

[3]Although standard MLM (*i.e.*, RoBERTa) does not have the decoder, its fine-tuning results are almost the same as $\delta = 1$ (with the decoder) as shown in Table 2.

2019) is commonly used to pretrain generative models that excel in text generation; MLM (Devlin et al., 2019; Liu et al., 2019) is prominently used to pretrain bidirectional text encoders to achieve superior performance for language understanding; Other language modeling objectives (Lewis et al., 2020a; Raffel et al., 2019) are designed to build sequence-to-sequence models that serve as both text generators and text encoders. As one of the most prominent pretraining approaches, MLM has stimulated many follow-up developments for pretraining bidirectional encoders (Bao et al., 2020; Clark et al., 2020; Gong et al., 2023; He et al., 2021; Joshi et al., 2019; Lan et al., 2020; Liao et al., 2022; Meng et al., 2021; 2022; Sanh et al., 2019; Yang et al., 2019). Remarkably, the idea of MLM is highly generalizable to different domains (Bao et al., 2022; Dosovitskiy et al., 2021; Hou et al., 2022; Tong et al., 2022; Wang et al., 2022; Xie et al., 2022) and leads to developments of unified pretraining frameworks for different modalities (Baevski et al., 2023; 2022). Given the broad impact of MLM, our analyses of representation deficiency in MLM may provide insights for future developments of pretraining algorithms in various fields.

**Study of Pretrained Models' Representations.** The powerful language representations learned by pretrained models have driven a series of studies to understand how linguistic knowledge is acquired through pretraining. Previous work studying the token representations in pretrained encoders has found that deeper layers generate more contextualized token representations (Ethayarajh, 2019), and these representations encode syntax structures (Goldberg, 2019; Hewitt & Manning, 2019) and fine-grained word senses (Coenen et al., 2019), offering supporting evidence for the effectiveness of pretrained models in downstream tasks. The success of learning such linguistic patterns is usually attributed to the self-attention mechanism which automatically learns to extract useful features through pretraining (Clark et al., 2019). Furthermore, different types of linguistic information are shown to be represented in a hierarchical way from shallower to deeper layers, reflecting the traditional NLP pipeline (Tenney et al., 2019a;b). There have also been prior efforts that investigate the limitations of pretrained models' representations. It has been revealed that the contextualized embedding space learned by pretrained models is generally anisotropic (Cai et al., 2021; Li et al., 2020) and is subject to a degeneration problem that token representations tend to be distributed into a narrow cone (Gao et al., 2019). Gong et al. (2019) identify that self-attention in Transformers tends to assign higher weights to local tokens as well as the starting token, which motivates the design of a progressive stacking algorithm for efficient pretraining. In this work, we investigate a previously unknown issue regarding MLM-pretrained models' representations that hinders the model's expressiveness on input sequences without `[MASK]` tokens. Our findings contribute a new perspective to understanding the limitations of representations in pretrained models.

## 6 CONCLUSION

**Limitations.** The focus of our work is on MLM and our analyses do not apply to other pretraining settings not using `[MASK]` tokens, and we discuss potential implications of our findings on autoregressive language models in Appendix F. While the current large language models are mostly autoregressive models, we believe that text encoder models still have important and wide applications in NLP, including but not limited to (1) Non-generation tasks. Many natural language understanding tasks do not have to be modeled autoregressively, for which encoder-only models are generally more parameter efficient and effective (Zhong et al., 2023). (2) Retrieval-augmented text generation (Lewis et al., 2020b), which typically uses an encoder for retrieval to enhance the generator's factualness. (3) Reward models in reinforcement learning from human feedback (RLHF) can use encoder models (Song et al., 2023). Empirically, we mainly compare with models pretrained by MLM and its simple variants and do not include all state-of-the-art models, as they typically require integrating multiple pretraining strategies and/or architecture changes (He et al., 2023).

**Conclusion.** In this work, we investigate the discrepancy caused by `[MASK]` tokens in MLM pretraining and demonstrate for the first time that this will necessarily result in real token representations being rank-deficient, thus limiting the model's expressiveness on real data without `[MASK]`. We propose a simple method MAE-LM that excludes `[MASK]` tokens from the encoder in pretraining to address the representation deficiency issue. We empirically show that MAE-LM improves the utilization of model dimensions for representing real tokens in pretraining and downstream tasks. MAE-LM consistently outperforms MLM-pretrained models on the GLUE and SQuAD benchmarks across multiple pretraining settings.

ACKNOWLEDGMENTS

Research was supported in part by U.S. National Science Foundation IIS-19-56151, the Molecule Maker Lab Institute: An AI Research Institutes program supported by NSF under Award No. 2019897, and the Institute for Geospatial Understanding through an Integrative Discovery Environment (I-GUIDE) by NSF under Award No. 2118329. Yu Meng was supported by a Google PhD Fellowship.

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

## A  DETAILED PROOFS

**Theorem 2.2** (Rank deficiency of real token representations). *There exists some layer $l$ in the Transformer encoder where the real token representation $\boldsymbol{H}^l_{\mathcal{R}}$ is rank-deficient. In particular, the row space of $\boldsymbol{H}^l_{\mathcal{R}}$ does not contain the row space of* `[MASK]` *token representation $\boldsymbol{H}^l_{\mathcal{M}}$.*

*Proof.* We prove the statement by contradiction: We suppose that the row space of $\boldsymbol{H}^l_{\mathcal{R}}$ always contains the row space of $\boldsymbol{H}^l_{\mathcal{M}}$ in all layers $1 \leq l \leq L$, and we will show that under this assumption, $\boldsymbol{H}^l_{\mathcal{M}}$ will converge exponentially (with $l$) to a rank-1 matrix, which contradicts with Lemma 2.1. In the following, we assume *single-head* self-attention is used, and the analysis can be easily generalized to the multi-head case.

The following proof extends Dong et al. (2021) by considering the representations of real tokens and mask tokens separately and following the residual norm analysis in Dong et al. (2021) to study the rank changes.

The self-attention module in the $l$th layer takes the previous layer representations $\boldsymbol{H}$ (the superscript $l - 1$ is omitted for convenience) as input and derives the output representations $\boldsymbol{H}'$:

$$
\begin{aligned}
\boldsymbol{H}' &= \mathrm{Attn}\left(\boldsymbol{H}\boldsymbol{W}^Q, \boldsymbol{H}\boldsymbol{W}^K, \boldsymbol{H}\boldsymbol{W}^V\right)\boldsymbol{W}^O \\
&= \mathrm{Softmax}\left(\frac{\boldsymbol{H}\boldsymbol{W}^Q\boldsymbol{W}^{K^\top}\boldsymbol{H}^\top}{\sqrt{d}}\right)\boldsymbol{H}\boldsymbol{W}^V\boldsymbol{W}^O \\
&= \boldsymbol{A}\boldsymbol{H}\boldsymbol{W}^{VO},
\end{aligned}
$$

where we denote the attention matrix computed from softmax as $\boldsymbol{A}$, and $\boldsymbol{W}^{VO} = \boldsymbol{W}^V\boldsymbol{W}^O$.

We study how the real token representations change (*i.e.*, comparing $\boldsymbol{H}'_{\mathcal{R}}$ with $\boldsymbol{H}_{\mathcal{R}}$) through the self-attention module. To facilitate easy analyses, we partition the input token representation matrix $\boldsymbol{H} \in \mathbb{R}^{(n+m) \times d}$ into blocks consisting of real token representations $\boldsymbol{H}_{\mathcal{R}} \in \mathbb{R}^{n \times d}$ and `[MASK]` token representations $\boldsymbol{H}_{\mathcal{M}} \in \mathbb{R}^{m \times d}$, and partition the attention matrix $\boldsymbol{A}_{\mathcal{R}}$ into blocks consisting of attention weights from real tokens to real tokens $\boldsymbol{A}_{\mathcal{R}:\mathcal{R}} \in \mathbb{R}^{n \times n}$ and from real tokens to `[MASK]` tokens $\boldsymbol{A}_{\mathcal{R}:\mathcal{M}} \in \mathbb{R}^{n \times m}$:

$$
\boldsymbol{H} = \begin{bmatrix} \boldsymbol{H}_{\mathcal{R}} \\ \boldsymbol{H}_{\mathcal{M}} \end{bmatrix}, \quad \boldsymbol{A}_{\mathcal{R}} = \begin{bmatrix} \boldsymbol{A}_{\mathcal{R}:\mathcal{R}} & \boldsymbol{A}_{\mathcal{R}:\mathcal{M}} \end{bmatrix}.
$$

We further denote

$$
\boldsymbol{S}_{\mathcal{R}:\mathcal{R}} = \exp\left[\boldsymbol{H}_{\mathcal{R}}\boldsymbol{W}^{QK}\boldsymbol{H}_{\mathcal{R}}^\top\right], \quad \boldsymbol{S}_{\mathcal{R}:\mathcal{M}} = \exp\left[\boldsymbol{H}_{\mathcal{R}}\boldsymbol{W}^{QK}\boldsymbol{H}_{\mathcal{M}}^\top\right], \quad \boldsymbol{Z} = \mathrm{diag}(\boldsymbol{S}_{\mathcal{R}:\mathcal{R}}\mathbf{1} + \boldsymbol{S}_{\mathcal{R}:\mathcal{M}}\mathbf{1}),
$$

where $\exp[\cdot]$ denotes the element-wise exponential function, $\mathrm{diag}(\cdot)$ constructs a diagnal matrix from a vector, $\boldsymbol{W}^{QK} = \boldsymbol{W}^Q\boldsymbol{W}^{K^\top}/\sqrt{d}$, and $\mathbf{1}$ is a vector of all ones. Then

$$
\boldsymbol{A}_{\mathcal{R}:\mathcal{R}} = \boldsymbol{Z}^{-1}\boldsymbol{S}_{\mathcal{R}:\mathcal{R}}, \quad \boldsymbol{A}_{\mathcal{R}:\mathcal{M}} = \boldsymbol{Z}^{-1}\boldsymbol{S}_{\mathcal{R}:\mathcal{M}}.
$$

Based on the above notations, the output representations at real token positions $\boldsymbol{H}'_{\mathcal{R}}$ can be written as:

$$
\boldsymbol{H}'_{\mathcal{R}} = \boldsymbol{A}_{\mathcal{R}}\boldsymbol{H}\boldsymbol{W}^{VO} = \begin{bmatrix} \boldsymbol{A}_{\mathcal{R}:\mathcal{R}} & \boldsymbol{A}_{\mathcal{R}:\mathcal{M}} \end{bmatrix} \begin{bmatrix} \boldsymbol{H}_{\mathcal{R}} \\ \boldsymbol{H}_{\mathcal{M}} \end{bmatrix} \boldsymbol{W}^{VO} = \boldsymbol{Z}^{-1}\left(\boldsymbol{S}_{\mathcal{R}:\mathcal{R}}\boldsymbol{H}_{\mathcal{R}} + \boldsymbol{S}_{\mathcal{R}:\mathcal{M}}\boldsymbol{H}_{\mathcal{M}}\right)\boldsymbol{W}^{VO}.
$$

$$(4)$$

If the row space of $\boldsymbol{H}_{\mathcal{R}}$ contains the row space of $\boldsymbol{H}_{\mathcal{M}}$, each row of $\boldsymbol{H}_{\mathcal{M}}$ can be represented as a linear combination of the rows in $\boldsymbol{H}_{\mathcal{R}}$:

$$
\boldsymbol{H}_{\mathcal{M}} = \boldsymbol{U}\boldsymbol{H}_{\mathcal{R}},
$$

where $\boldsymbol{U} \in \mathbb{R}^{m \times n}$ is the linear combination weight matrix. We can rescale the vector norm of each row in $\boldsymbol{H}_{\mathcal{M}}$ so that $\boldsymbol{U}$ has a row sum of one (*i.e.*, $\boldsymbol{U}\mathbf{1} = \mathbf{1}$).

To examine the rank of real token representations, we examine the change in matrix residual through Transformer layers, inspired by Dong et al. (2021). Specifically, we define the following residual $\boldsymbol{R}$ which measures the difference between $\boldsymbol{H}_{\mathcal{R}}$ and a rank-1 matrix:

$$
\boldsymbol{R} = \boldsymbol{H}_{\mathcal{R}} - \mathbf{1}\boldsymbol{h}^\top, \quad \boldsymbol{h} = \arg\min_{\boldsymbol{x}} \left\|\boldsymbol{H}_{\mathcal{R}} - \mathbf{1}\boldsymbol{x}^\top\right\|.
$$

We aim to show that the norm of $\boldsymbol{R}$ converges exponentially (with layer depth) to zero, meaning that $\boldsymbol{H}_{\mathcal{R}}$ converges (with layer depth) to a rank-1 matrix.

By plugging $\boldsymbol{H}_{\mathcal{R}} = \boldsymbol{R} + \boldsymbol{1}\boldsymbol{h}^{\top}$ and $\boldsymbol{H}_{\mathcal{M}} = \boldsymbol{U}\boldsymbol{H}_{\mathcal{R}} = \boldsymbol{U}\boldsymbol{R} + \boldsymbol{U}\boldsymbol{1}\boldsymbol{h}^{\top} = \boldsymbol{U}\boldsymbol{R} + \boldsymbol{1}\boldsymbol{h}^{\top}$ into Equation (4), we obtain

$$
\begin{aligned}
\boldsymbol{H}'_{\mathcal{R}} &= \boldsymbol{Z}^{-1}\left(\boldsymbol{S}_{\mathcal{R}:\mathcal{R}}\left(\boldsymbol{R} + \boldsymbol{1}\boldsymbol{h}^{\top}\right) + \boldsymbol{S}_{\mathcal{R}:\mathcal{M}}\left(\boldsymbol{U}\boldsymbol{R} + \boldsymbol{1}\boldsymbol{h}^{\top}\right)\right)\boldsymbol{W}^{VO} \\
&= \left(\boldsymbol{Z}^{-1}\left(\boldsymbol{S}_{\mathcal{R}:\mathcal{R}} + \boldsymbol{S}_{\mathcal{R}:\mathcal{M}}\boldsymbol{U}\right)\boldsymbol{R} + \underbrace{\boldsymbol{Z}^{-1}\left(\boldsymbol{S}_{\mathcal{R}:\mathcal{R}}\boldsymbol{1} + \boldsymbol{S}_{\mathcal{R}:\mathcal{M}}\boldsymbol{1}\right)}_{=\boldsymbol{1}}\boldsymbol{h}^{\top}\right)\boldsymbol{W}^{VO} \\
&= \boldsymbol{Z}^{-1}\left(\boldsymbol{S}_{\mathcal{R}:\mathcal{R}} + \boldsymbol{S}_{\mathcal{R}:\mathcal{M}}\boldsymbol{U}\right)\boldsymbol{R}\boldsymbol{W}^{VO} + \boldsymbol{1}\boldsymbol{h}^{\top}\boldsymbol{W}^{VO}. \quad (5)
\end{aligned}
$$

Next we write out $\boldsymbol{S}_{\mathcal{R}:\mathcal{R}}$ and $\boldsymbol{S}_{\mathcal{R}:\mathcal{M}}$:

$$
\begin{aligned}
\boldsymbol{S}_{\mathcal{R}:\mathcal{R}} &= \exp\left[\boldsymbol{H}_{\mathcal{R}}\boldsymbol{W}^{QK}\boldsymbol{H}_{\mathcal{R}}^{\top}\right] \\
&= \exp\left[(\boldsymbol{R} + \boldsymbol{1}\boldsymbol{h}^{\top})\boldsymbol{W}^{QK}(\boldsymbol{R} + \boldsymbol{1}\boldsymbol{h}^{\top})^{\top}\right] \\
&= \exp\left[\boldsymbol{R}\boldsymbol{W}^{QK}\boldsymbol{R}^{\top} + \boldsymbol{1}\boldsymbol{h}^{\top}\boldsymbol{W}^{QK}\boldsymbol{R}^{\top} + \left(\boldsymbol{R}\boldsymbol{W}^{QK}\boldsymbol{h} + \boldsymbol{1}\boldsymbol{h}^{\top}\boldsymbol{W}^{QK}\boldsymbol{h}\right)\boldsymbol{1}^{\top}\right] \\
&= \exp\left[\underbrace{\boldsymbol{R}\boldsymbol{W}^{QK}\boldsymbol{R}^{\top}}_{=\boldsymbol{F}}\right] \odot \exp\left[\boldsymbol{1}\underbrace{\boldsymbol{h}^{\top}\boldsymbol{W}^{QK}\boldsymbol{R}^{\top}}_{=\boldsymbol{g}^{\top}}\right] \odot \exp\left[\underbrace{\left(\boldsymbol{R}\boldsymbol{W}^{QK}\boldsymbol{h} + \boldsymbol{1}\boldsymbol{h}^{\top}\boldsymbol{W}^{QK}\boldsymbol{h}\right)}_{=\boldsymbol{c}}\boldsymbol{1}^{\top}\right],
\end{aligned}
$$

and

$$
\begin{aligned}
\boldsymbol{S}_{\mathcal{R}:\mathcal{M}} &= \exp\left[\boldsymbol{H}_{\mathcal{R}}\boldsymbol{W}^{QK}\boldsymbol{H}_{\mathcal{M}}^{\top}\right] \\
&= \exp\left[(\boldsymbol{R} + \boldsymbol{1}\boldsymbol{h}^{\top})\boldsymbol{W}^{QK}(\boldsymbol{U}\boldsymbol{R} + \boldsymbol{1}\boldsymbol{h}^{\top})^{\top}\right] \\
&= \exp\left[\boldsymbol{R}\boldsymbol{W}^{QK}\boldsymbol{R}^{\top}\boldsymbol{U}^{\top} + \boldsymbol{1}\boldsymbol{h}^{\top}\boldsymbol{W}^{QK}\boldsymbol{R}^{\top}\boldsymbol{U}^{\top} + \left(\boldsymbol{R}\boldsymbol{W}^{QK}\boldsymbol{h} + \boldsymbol{1}\boldsymbol{h}^{\top}\boldsymbol{W}^{QK}\boldsymbol{h}\right)\boldsymbol{1}^{\top}\right] \\
&= \exp\left[\underbrace{\boldsymbol{R}\boldsymbol{W}^{QK}\boldsymbol{R}^{\top}\boldsymbol{U}^{\top}}_{=\boldsymbol{F}'}\right] \odot \exp\left[\boldsymbol{1}\underbrace{\boldsymbol{h}^{\top}\boldsymbol{W}^{QK}\boldsymbol{R}^{\top}\boldsymbol{U}^{\top}}_{=\boldsymbol{g}'^{\top}}\right] \odot \exp\left[\underbrace{\left(\boldsymbol{R}\boldsymbol{W}^{QK}\boldsymbol{h} + \boldsymbol{1}\boldsymbol{h}^{\top}\boldsymbol{W}^{QK}\boldsymbol{h}\right)}_{=\boldsymbol{c}}\boldsymbol{1}^{\top}\right],
\end{aligned}
$$

where $\odot$ denotes the element-wise product. Let $\boldsymbol{F} = \boldsymbol{R}\boldsymbol{W}^{QK}\boldsymbol{R}^{\top}$, $\boldsymbol{F}' = \boldsymbol{R}\boldsymbol{W}^{QK}\boldsymbol{R}^{\top}\boldsymbol{U}^{\top}$, $\boldsymbol{g}^{\top} = \boldsymbol{h}^{\top}\boldsymbol{W}^{QK}\boldsymbol{R}^{\top}$, $\boldsymbol{g}'^{\top} = \boldsymbol{h}^{\top}\boldsymbol{W}^{QK}\boldsymbol{R}^{\top}\boldsymbol{U}^{\top}$, and $\boldsymbol{c} = \boldsymbol{R}\boldsymbol{W}^{QK}\boldsymbol{h} + \boldsymbol{1}\boldsymbol{h}^{\top}\boldsymbol{W}^{QK}\boldsymbol{h}$, we can further write out $\boldsymbol{Z}$:

$$
\begin{aligned}
\boldsymbol{Z} &= \operatorname{diag}\left(\boldsymbol{S}_{\mathcal{R}:\mathcal{R}}\boldsymbol{1} + \boldsymbol{S}_{\mathcal{R}:\mathcal{M}}\boldsymbol{1}\right) \\
&= \operatorname{diag}\left(\left(\left(\exp\left[\boldsymbol{F}\right] \odot \exp\left[\boldsymbol{1}\boldsymbol{g}^{\top}\right]\right)\boldsymbol{1} + \left(\exp\left[\boldsymbol{F}'\right] \odot \exp\left[\boldsymbol{1}\boldsymbol{g}'^{\top}\right]\right)\boldsymbol{1}\right) \odot \exp\left[\boldsymbol{c}\right]\right).
\end{aligned}
$$

Let $\widetilde{\boldsymbol{F}} = [\boldsymbol{F} \quad \boldsymbol{F}']$ be the augmented matrix by combining the columns of $\boldsymbol{F}$ and $\boldsymbol{F}'$, and let $\overline{\boldsymbol{f}}$ and $\underline{\boldsymbol{f}}$ denote the maximum and minimum element across each row of $\widetilde{\boldsymbol{F}}$, respectively:

$$
\overline{f}_i = \max_j \widetilde{F}_{ij}, \quad \underline{f}_i = \min_j \widetilde{F}_{ij}.
$$

Then we can derive a lower bound of each element in $\boldsymbol{Z}^{-1}\boldsymbol{S}_{\mathcal{R}:\mathcal{R}}$:

$$
\begin{aligned}
\left[\boldsymbol{Z}^{-1}\boldsymbol{S}_{\mathcal{R}:\mathcal{R}}\right]_{ij} &= \frac{\exp(F_{ij})\exp(g_j)\exp(c_i)}{\left(\sum_{j'}\exp(F_{ij'})\exp(g_{j'}) + \sum_{j'}\exp(F'_{ij'})\exp(g'_{j'})\right)\exp(c_i)} \\
&\geq \frac{\exp(F_{ij})\exp(g_j)}{\exp\left(\overline{f}_i\right)\left(\sum_{j'}\exp(g_{j'}) + \sum_{j'}\exp(g'_{j'})\right)} \\
&= \exp\left(F_{ij} - \overline{f}_i\right)\frac{\exp(g_j)}{\sum_{j'}\exp(g_{j'}) + \sum_{j'}\exp(g'_{j'})}.
\end{aligned}
$$

Similarly, we can derive an upper bound:

$$
\left[\boldsymbol{Z}^{-1}\boldsymbol{S}_{\mathcal{R}:\mathcal{R}}\right]_{ij} \leq \exp\left(F_{ij} - \underline{f}_i\right)\frac{\exp(g_j)}{\sum_{j'}\exp(g_{j'}) + \sum_{j'}\exp(g'_{j'})}.
$$

Using the the Taylor expansion of exp, we have

$$\exp\left(F_{ij} - \overline{f}_i\right) \geq 1 + F_{ij} - \overline{f}_i \geq 1 + \underline{f}_i - \overline{f}_i, \quad \exp\left(F_{ij} - \underline{f}_i\right) \leq 1 + 2\left(F_{ij} - \underline{f}_i\right) \leq 1 + 2\left(\overline{f}_i - \underline{f}_i\right).$$

Therefore,

$$(1 + \underline{f}_i - \overline{f}_i)\frac{\exp(g_j)}{\sum_{j'}\exp(g_{j'}) + \sum_{j'}\exp(g'_{j'})} \leq \left[\boldsymbol{Z}^{-1}\boldsymbol{S}_{\mathcal{R}:\mathcal{R}}\right]_{ij} \leq (1 + 2\overline{f}_i - 2\underline{f}_i)\frac{\exp(g_j)}{\sum_{j'}\exp(g_{j'}) + \sum_{j'}\exp(g'_{j'})}.$$

Denote $\boldsymbol{D} = \text{diag}\left(\overline{\boldsymbol{f}} - \underline{\boldsymbol{f}}\right)$ and $g_+ = \exp\left[\boldsymbol{g}^\top\right]\mathbf{1} + \exp\left[\boldsymbol{g}'^\top\right]\mathbf{1}$, then the above bound can be expressed in matrix form as follows (the inequality between matrices holds element-wise):

$$\frac{1}{g_+}(\boldsymbol{I} - \boldsymbol{D})\mathbf{1}\exp\left[\boldsymbol{g}^\top\right] \leq \boldsymbol{Z}^{-1}\boldsymbol{S}_{\mathcal{R}:\mathcal{R}} \leq \frac{1}{g_+}(\boldsymbol{I} + 2\boldsymbol{D})\mathbf{1}\exp\left[\boldsymbol{g}^\top\right]. \tag{6}$$

An analogous derivation gives the bound of $\boldsymbol{Z}^{-1}\boldsymbol{S}_{\mathcal{R}:\mathcal{M}}$:

$$\frac{1}{g_+}(\boldsymbol{I} - \boldsymbol{D})\mathbf{1}\exp\left[\boldsymbol{g}'^\top\right] \leq \boldsymbol{Z}^{-1}\boldsymbol{S}_{\mathcal{R}:\mathcal{M}} \leq \frac{1}{g_+}(\boldsymbol{I} + 2\boldsymbol{D})\mathbf{1}\exp\left[\boldsymbol{g}'^\top\right]. \tag{7}$$

Since the upper and lower bounds are in very similar forms, we will only focus on the upper bound in the derivations below.

Combining Equation (6) with Equation (7), we have

$$\boldsymbol{Z}^{-1}\left(\boldsymbol{S}_{\mathcal{R}:\mathcal{R}} + \boldsymbol{S}_{\mathcal{R}:\mathcal{M}}\boldsymbol{U}\right) \leq \mathbf{1}\underbrace{\left(\frac{\exp\left[\boldsymbol{g}^\top\right] + \exp\left[\boldsymbol{g}'^\top\right]\boldsymbol{U}}{g_+}\right)}_{=\boldsymbol{r}^\top} + 2\boldsymbol{D}\mathbf{1}\underbrace{\left(\frac{\exp\left[\boldsymbol{g}^\top\right] + \exp\left[\boldsymbol{g}'^\top\right]\boldsymbol{U}}{g_+}\right)}_{=\boldsymbol{r}^\top}$$
$$= \mathbf{1}\boldsymbol{r}^\top + 2\boldsymbol{D}\mathbf{1}\boldsymbol{r}^\top \tag{8}$$

Plugging Equation (8) into Equation (5), we have

$$\boldsymbol{H}'_{\mathcal{R}} \leq \left(\mathbf{1}\boldsymbol{r}^\top + 2\boldsymbol{D}\mathbf{1}\boldsymbol{r}^\top\right)\boldsymbol{R}\boldsymbol{W}^{VO} + \mathbf{1}\boldsymbol{h}^\top\boldsymbol{W}^{VO} = \mathbf{1}\underbrace{\left(\boldsymbol{r}^\top\boldsymbol{R}\boldsymbol{W}^{VO} + \boldsymbol{h}^\top\boldsymbol{W}^{VO}\right)}_{=\boldsymbol{h}'^\top} + 2\boldsymbol{D}\mathbf{1}\boldsymbol{r}^\top\boldsymbol{R}\boldsymbol{W}^{VO}.$$

Therefore,

$$\boldsymbol{H}'_{\mathcal{R}} - \mathbf{1}\boldsymbol{h}'^\top \leq 2\boldsymbol{D}\mathbf{1}\boldsymbol{r}^\top\boldsymbol{R}\boldsymbol{W}^{VO}.$$

With a similar derivation, we have the following lower bound:

$$\boldsymbol{H}'_{\mathcal{R}} - \mathbf{1}\boldsymbol{h}'^\top \geq -\boldsymbol{D}\mathbf{1}\boldsymbol{r}^\top\boldsymbol{R}\boldsymbol{W}^{VO}.$$

Overall, we can bound the element-wise absolute values of $\boldsymbol{R}' = \boldsymbol{H}'_{\mathcal{R}} - \mathbf{1}\boldsymbol{h}'^\top$, which measure the distance between $\boldsymbol{H}'_{\mathcal{R}}$ and a rank-1 matrix:

$$\left|R'_{ij}\right| = \left|\left[\boldsymbol{H}'_{\mathcal{R}} - \mathbf{1}\boldsymbol{h}'^\top\right]_{ij}\right| \leq \left|\left[2\boldsymbol{D}\mathbf{1}\boldsymbol{r}^\top\boldsymbol{R}\boldsymbol{W}^{VO}\right]_{ij}\right|.$$

This allows us to further bound the norm of $\boldsymbol{R}'$. For $\ell_1$ norm, we have

$$\begin{aligned}
\left\|\boldsymbol{R}'\right\|_1 &\leq \left\|2\boldsymbol{D}\mathbf{1}\boldsymbol{r}^\top\boldsymbol{R}\boldsymbol{W}^{VO}\right\|_1 \\
&\leq 2\left\|\boldsymbol{D}\mathbf{1}\right\|_\infty\left\|\boldsymbol{r}^\top\boldsymbol{R}\boldsymbol{W}^{VO}\right\|_1 && \text{Based on Hölder's inequality} \\
&\leq 2\left\|\boldsymbol{D}\mathbf{1}\right\|_\infty\left\|\boldsymbol{r}^\top\right\|_1\left\|\boldsymbol{R}\right\|_1\left\|\boldsymbol{W}^{VO}\right\|_1, && \text{Submultiplicativity of matrix norms}
\end{aligned}$$

where

$$\begin{aligned}
\left\|\boldsymbol{D}\mathbf{1}\right\|_\infty &= \max_i\left|\overline{f}_i - \underline{f}_i\right| \\
&\leq 2\left\|\widetilde{\boldsymbol{F}}\right\|_1 \\
&\leq 2\max\left\{\left\|\boldsymbol{R}\boldsymbol{W}^{QK}\boldsymbol{R}^\top\right\|_1, \left\|\boldsymbol{R}\boldsymbol{W}^{QK}\boldsymbol{R}^\top\boldsymbol{U}^\top\right\|_1\right\} \\
&\leq 2\left\|\boldsymbol{R}\right\|_1\left\|\boldsymbol{W}^{QK}\right\|_1\left\|\boldsymbol{R}\right\|_\infty\max\left\{1, \left\|\boldsymbol{U}\right\|_\infty\right\} \\
&\leq 2\left\|\boldsymbol{R}\right\|_1\left\|\boldsymbol{W}^{QK}\right\|_1\left\|\boldsymbol{R}\right\|_\infty\left\|\boldsymbol{U}\right\|_\infty, && \left\|\boldsymbol{U}\right\|_\infty \geq 1 \text{ since } \boldsymbol{U}\mathbf{1} = \mathbf{1}
\end{aligned}$$

and

$$\left\|\boldsymbol{r}^\top\right\|_1 \le \left\|\boldsymbol{r}^\top\right\|_\infty$$
$$= \left\|\frac{\exp\left[\boldsymbol{g}^\top\right] + \exp\left[\boldsymbol{g}'^\top\right]\boldsymbol{U}}{g_+}\right\|_\infty$$
$$\le \left\|\frac{\exp\left[\boldsymbol{g}^\top\right]}{g_+}\right\|_\infty + \left\|\frac{\exp\left[\boldsymbol{g}'^\top\right]\boldsymbol{U}}{g_+}\right\|_\infty$$
$$\le 1 + \|\boldsymbol{U}\|_\infty .$$

Therefore, we can bound the $\ell_1$ norm of $\|\boldsymbol{R}'\|_1$ as follows:

$$\|\boldsymbol{R}'\|_1 \le 4 \left\|\boldsymbol{W}^{QK}\right\|_1 \left\|\boldsymbol{W}^{VO}\right\|_1 \|\boldsymbol{U}\|_\infty \left(1 + \|\boldsymbol{U}\|_\infty\right) \|\boldsymbol{R}\|_1^2 \|\boldsymbol{R}\|_\infty . \tag{9}$$

Similarly, we can obtain the bound for the $\ell_\infty$ norm of $\|\boldsymbol{R}'\|_1$:

$$\|\boldsymbol{R}'\|_\infty \le 4 \left\|\boldsymbol{W}^{QK}\right\|_1 \left\|\boldsymbol{W}^{VO}\right\|_\infty \|\boldsymbol{U}\|_\infty \left(1 + \|\boldsymbol{U}\|_\infty\right) \|\boldsymbol{R}\|_1 \|\boldsymbol{R}\|_\infty^2 . \tag{10}$$

Denote the geometric mean of $\|\boldsymbol{R}\|_1$ and $\|\boldsymbol{R}\|_\infty$ as $\|\boldsymbol{R}\|_{1,\infty} = \sqrt{\|\boldsymbol{R}\|_1 \|\boldsymbol{R}\|_\infty}$, then from Equation (9) and Equation (10), we have

$$\|\boldsymbol{R}'\|_{1,\infty} \le 4 \underbrace{\left\|\boldsymbol{W}^{QK}\right\|_1 \left\|\boldsymbol{W}^{VO}\right\|_{1,\infty} \|\boldsymbol{U}\|_\infty \left(1 + \|\boldsymbol{U}\|_\infty\right)}_{=\epsilon} \|\boldsymbol{R}\|_{1,\infty}^3$$
$$= 4\epsilon \|\boldsymbol{R}\|_{1,\infty}^3 .$$

The above inequality reflects how the residual changes within one self-attention layer. Applying it recursively throughout all layers in an $L$-layer encoder, we have:

$$\left\|\boldsymbol{R}^L\right\|_{1,\infty} \le (4\bar{\epsilon})^{\frac{3^L-1}{2}} \left\|\boldsymbol{R}^0\right\|_{1,\infty}^{3^L} , \quad \bar{\epsilon} = \max_l \epsilon^l,$$

where $\boldsymbol{R}^L$ and $\boldsymbol{R}^0$ denote the residuals corresponding to the encoder's output real token representations $\boldsymbol{H}_\mathcal{R}^L$ and input real token representations $\boldsymbol{H}_\mathcal{R}^0$, respectively.

This demonstrates that the residual norms of real token representations converge exponentially (with layer depth) to zero. Hence, the real token representation matrix $\boldsymbol{H}_\mathcal{R}^l$ converges exponentially (with layer depth) to a rank-1 matrix. Since the row space of [MASK] token representations $\boldsymbol{H}_\mathcal{M}^l$ is contained by the row space of $\boldsymbol{H}_\mathcal{R}^l$, we have $\text{rank}(\boldsymbol{H}_\mathcal{M}^l) \le \text{rank}(\boldsymbol{H}_\mathcal{R}^l)$, and $\boldsymbol{H}_\mathcal{M}^l$ will also converge exponentially (with layer depth) to a rank-1 matrix, which contradicts with Lemma 2.1. Finally, we conclude that the row space of $\boldsymbol{H}_\mathcal{R}^l$ must not contain the row space of $\boldsymbol{H}_\mathcal{M}^l$, which necessarily implies that $\boldsymbol{H}_\mathcal{R}^l$ is rank-deficient. □

## B DETAILS ABOUT GLUE TASKS

More details of all the GLUE tasks can be found as follows.

**MNLI:** The Multi-genre Natural Language Inference (Williams et al., 2018) task includes 393K training examples from crowdsourcing. The goal is to predict if a premise sentence entails, contradicts, or is neutral with respect to a given hypothesis sentence.

**QQP:** Question Pairs (Shankar et al., 2017) includes 364K training examples from the Quora question-answering website. The task is to determine if two given questions are semantically equivalent.

**QNLI:** Question Natural Language Inference includes 108K training examples derived from the Stanford Question Answering Dataset (SQuAD) (Rajpurkar et al., 2018). The task is to predict if a sentence contains the answer to a given question.

**SST-2:** Stanford Sentiment Treebank (Socher et al., 2013) includes 67K training examples on movie reviews with human annotations. The task is to determine if a given sentence has positive or negative sentiment.

**CoLA:** Corpus of Linguistic Acceptability (Warstadt et al., 2019) includes 8.5K training examples from books and journal articles on linguistic theory. The task is to determine if a given sentence is linguistically acceptable.

**RTE:** Recognizing Textual Entailment (Bentivogli et al., 2009; Dagan et al., 2005; Haim et al., 2006; Giampiccolo et al., 2007) includes 2.5K training examples from textual entailment challenges. The task is to predict if a premise sentence entails a given hypothesis sentence.

**MRPC:** Microsoft Research Paraphrase Corpus (Dolan & Brockett, 2005) includes 3.7K training examples collected from news sources. The task is to predict if two given sentences are semantically equivalent.

**STS-B:** Semantic Textual Similarity (Cer et al., 2017) includes 5.8K training examples collected from multiple sources on sentence pair semantic similarity annotated by humans. The task is to predict the semantic similarity of two sentences (based on a 1 to 5 scoring scale).

## C  IMPLEMENTATION DETAILS

**Details of Pretraining Settings.** The *base* setting follows $BERT_{base}$ (Devlin et al., 2019) pretraining which uses Wikipedia and BookCorpus (Zhu et al., 2015) (16GB of texts) as the pretraining corpora. The encoder architecture is a 12-layer Transformer, and the model dimension is 768. We train both absolute and relative position embeddings (Raffel et al., 2019) in the encoder. The decoder is a 4-layer Transformer with the same model dimensions as the encoder. Since the decoder is not used in downstream tasks, MAE-LM's encoder can be fairly compared with previous 12-layer base-sized models. The model is trained for 125K steps with $2,048$ sequences per batch, which amounts to 256M samples in total. The maximum input sequence length is 512 tokens. The vocabulary is constructed with BPE (Sennrich et al., 2015) and consists of $32,768$ *uncased* subword units.

The *base++* setting follows RoBERTa (Liu et al., 2019) pretraining which extends the *base* setting by incorporating larger pretraining corpora and training the same model architecture for longer. Specifically, the following corpora are used along with Wikipedia and BookCorpus: OpenWeb-Text (Gokaslan & Cohen, 2019), CC-News (Liu et al., 2019), and STORIES (Trinh & Le, 2018). This expands the pretraining corpora to contain 160GB texts. The model is trained for 2M steps with $2,048$ sequences per batch, which amounts to 4B samples in total. The *base++* setting also expands the vocabulary size to $64,000$ (Bao et al., 2020) by using *cased* subword units.

The *large++* setting extends the *base++* setting by scaling up the encoder architecture to 24 layers and $1,024$ model dimensions. The decoder is still a 4-layer Transformer with the same model dimensions as the encoder. Due to the high cost of training large models, we train for 1M steps (half of the *base++* setting) with $2,048$ sequences per batch, which amounts to 2B samples in total. Note that this is also half of the pretraining data used in RoBERTa (Liu et al., 2019) and BART (Lewis et al., 2020a).

**Computation Environment.** The experiments in this paper are conducted on 64 A100 GPUs.

**Masking.** For all pretraining settings, we apply $15\%$ random masks to input sequeces. We do not use the trick in conventional MLM (Devlin et al., 2019; Liu et al., 2019) that replaces $10\%$ of `[MASK]` tokens with the original ones and another $10\%$ with random tokens. We also experiment with higher masking rates (*e.g.*, $40\%$) which are shown to be beneficial in Wettig et al. (2023) for training large models, but they do not yield better results than the default $15\%$ masking rate in our experiments. This is probably because Wettig et al. (2023) use an efficient pretraining recipe that is different from the standard pretraining setup, with a larger learning rate, a larger batch size, a shorter sequence length, and fewer training steps.

**Position Embedding.** We learn both absolute and relative position embeddings (Raffel et al., 2019) in the encoder, and only learn absolute position embeddings in the decoder.

**Dropout.** During the pretraining of MAE-LM, dropout is applied to the encoder but not the decoder, which we find to slightly improve stability.

## D  HYPERPARAMETER SETTINGS

Table 3: Hyperparameters used in pretraining.

| Hyperparameter | *base* | *base++* | *large++* |
|---|---|---|---|
| Max Steps | 125K | 2M | 1M |
| Peak Learning Rate | 5e-4 | 2e-4 | 1e-4 |
| Batch Size | 2048 | 2048 | 2048 |
| Warm-Up Steps | 10K | 10K | 10K |
| Sequence Length | 512 | 512 | 512 |
| Relative Position Encoding Buckets | 32 | 64 | 128 |
| Relative Position Encoding Max Distance | 128 | 128 | 256 |
| Adam $\epsilon$ | 1e-6 | 1e-6 | 1e-6 |
| Adam $(\beta_1, \beta_2)$ | (0.9, 0.98) | (0.9, 0.98) | (0.9, 0.98) |
| Clip Norm | 2.0 | 2.0 | 1.0 |
| Dropout | 0.1 | 0.1 | 0.1 |
| Weight Decay | 0.01 | 0.01 | 0.01 |

Table 4: Hyperparameter ranges searched for fine-tuning on GLUE. GLUE small tasks include CoLA, RTE, MRPC and STS-B. GLUE large tasks include MNLI, QQP, QNLI and SST-2.

| Hyperparameter | GLUE Small Tasks Search Space | GLUE Large Tasks Search Space |
|---|---|---|
| Max Epochs | {2, 3, 5, 10} | {2, 3, 5} |
| Peak Learning Rate | *base/base++*: {2e-5, 3e-5, 4e-5, 5e-5} *large++*: {7e-6, 1e-5, 2e-5, 3e-5} | *base/base++*: {1e-5, 2e-5, 3e-5, 4e-5} *large++*: {5e-6, 7e-6, 1e-5, 2e-5} |
| Batch Size | {16, 32} | 32 |
| Warm-Up Proportion | {6%, 10%} | 6% |
| Sequence Length | 512 | 512 |
| Adam $\epsilon$ | 1e-6 | 1e-6 |
| Adam $(\beta_1, \beta_2)$ | (0.9, 0.98) | (0.9, 0.98) |
| Clip Norm | - | - |
| Dropout | 0.1 | 0.1 |
| Weight Decay | 0.01 | 0.01 |

Table 5: Hyperparameter ranges searched for fine-tuning on SQuAD 2.0.

| Hyperparameter | SQuAD 2.0 Search Space |
|---|---|
| Max Epochs | {2, 3} |
| Peak Learning Rate | *base/base++*: {2e-5, 3e-5, 4e-5, 5e-5} *large++*: {7e-6, 1e-5, 2e-5, 3e-5} |
| Batch Size | {16, 32} |
| Warm-Up Proportion | {6%, 10%} |
| Sequence Length | 512 |
| Adam $\epsilon$ | 1e-6 |
| Adam $(\beta_1, \beta_2)$ | (0.9, 0.98) |
| Clip Norm | - |
| Dropout | 0.1 |
| Weight Decay | 0.01 |

Table 6: Standard single-task, single-model fine-tuning results (medians over five random seeds) evaluated on GLUE and SQuAD 2.0 development sets for large models. [†]: MAE-LM is pretrained on half of RoBERTa/BART's data.

| Model | GLUE (Single-Task) | | | | | | | | | SQuAD 2.0 | |
| | MNLI-(m/mm) | QQP | QNLI | SST-2 | CoLA | RTE | MRPC | STS-B | AVG | EM | F1 |
|---|---|---|---|---|---|---|---|---|---|---|---|
| | *large++* setting: larger Transformer model trained on larger pretraining corpora (160GB) | | | | | | | | | | |
| BART | 89.9/90.1 | **92.5** | 94.9 | **96.6** | 62.8 | 87.0 | 90.4 | 91.2 | 88.2 | 86.1 | 89.2 |
| RoBERTa | 90.2/90.2 | 92.2 | 94.7 | 96.4 | 68.0 | 86.6 | **90.9** | **92.4** | 88.9 | 86.5 | 89.4 |
| MAE-LM [†] | **90.4/90.6** | 92.2 | **95.1** | 96.2 | **68.7** | **88.8** | 90.7 | 92.1 | **89.3** | 87.0 | 89.8 |

Table 7: Zero-shot and few-shot performance. Few-shot results include mean and standard deviation (as subscripts) performance over 5 different training splits defined in Gao et al. (2021). [†]: Results from Gao et al. (2021).

| Model | GLUE (Single-Task) | | | | | | | | |
| | MNLI-(m/mm) | QQP | QNLI | SST-2 | CoLA | RTE | MRPC | STS-B | AVG |
|---|---|---|---|---|---|---|---|---|---|
| | *zero-shot prompting*: direct inference on tasks via cloze-type MLM predictions | | | | | | | | |
| RoBERTa[†] | 50.8/51.7 | 49.7 | 50.8 | 83.6 | 2.0 | 51.3 | 61.9 | −3.2 | 43.4 |
| MAE-LM | 52.1/54.3 | 52.0 | 52.3 | 83.5 | 2.0 | 54.5 | 63.4 | −3.0 | 44.7 |
| | *head-based few-shot fine-tuning*: fine-tuning on 16 samples per label with a linear classification head | | | | | | | | |
| RoBERTa[†] | $45.8_{6.4}/47.8_{6.8}$ | $60.7_{4.3}$ | $60.2_{6.5}$ | $81.4_{3.8}$ | $33.9_{14.3}$ | $54.4_{3.9}$ | $76.6_{2.5}$ | $53.5_{8.5}$ | 58.4 |
| MAE-LM | $48.7_{4.5}/51.1_{6.0}$ | $64.5_{4.2}$ | $62.1_{6.1}$ | $81.2_{3.9}$ | $31.1_{13.9}$ | $58.0_{2.5}$ | $78.2_{2.1}$ | $53.0_{9.0}$ | 59.8 |
| | *prompt-based few-shot fine-tuning*: fine-tuning on 16 samples per label with cloze-type MLM templates | | | | | | | | |
| RoBERTa[†] | $68.3_{2.3}/70.5_{1.9}$ | $65.5_{5.3}$ | $64.5_{4.2}$ | $92.7_{0.9}$ | $9.3_{7.3}$ | $69.1_{3.6}$ | $74.5_{5.3}$ | $71.0_{7.0}$ | 64.5 |
| MAE-LM | $70.7_{2.0}/73.3_{1.8}$ | $67.3_{4.6}$ | $65.1_{4.3}$ | $92.4_{1.1}$ | $14.3_{8.9}$ | $71.2_{3.3}$ | $74.8_{4.1}$ | $72.3_{6.5}$ | 66.2 |

We report the detailed hyperparameters used for pretraining in Table 3. The hyperparameter search ranges of fine-tuning are shown in Tables 4 and 5 for GLUE and SQuAD 2.0, respectively.

For fair comparisons, the same set of hyperparameters (in both pretraining and fine-tuning) is used for MAE-LM, RoBERTa (Ours) and ablations. We follow previous pretraining studies (Liu et al., 2019) to report the medians of downstream task fine-tuning results under the same set of five different random seeds.

# E   MORE EVALUATION RESULTS

**BERT Masking Strategy.** In addition to our default masking strategy which directly applies 15% random masks to input sequences, we also validate our findings under the original BERT masking strategy that replaces 10% of [MASK] tokens with the original ones and another 10% with random tokens. Figure 6 demonstrates that the gap in effective representation rank between inputs with and without [MASK] under this setting is also notable, similar to the findings in Figure 1(a). This confirms that randomly replacing a small percentage of [MASK] tokens with real tokens does not effectively address the representation deficiency issue, as the ratio of [MASK] tokens in pretraining is still high.

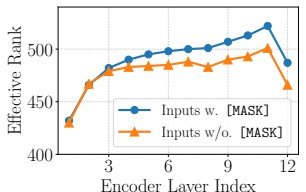

Figure 6: With the original BERT masking strategy, the effective rank across layers for inputs without [MASK] and with [MASK].

**Large Model Results.**   We also show the performance of MAE-LM under larger model sizes in Table 6. Even trained on half of the pretraining data used in RoBERTa (Liu et al., 2019), MAE-LM still performs comparably or better, demonstrating the potential of MAE-LM for larger models.

**Zero-Shot and Few-Shot Results.**   Since MAE-LM is trained with the MLM objective, it is applicable to zero-shot and few-shot learning via prompt-based approaches. We report three groups of zero-shot/few-shot results on the GLUE tasks comparing MAE-LM (*large++*) with RoBERTa

(*large++*) in Table 7: (1) *zero-shot prompting* which converts the classification tasks into cloze-type MLM predictions and directly uses pretrained models for inference on test sets; (2) *head-based few-shot fine-tuning* which adds a linear classification head to the pretrained encoders for fine-tuning on 16 samples per label; and (3) *few-shot prompt-based fine-tuning* which fine-tunes the MLM models on tasks converted to cloze-type MLM formats with 16 samples per label. We follow the basic manual prompt/label word setting and the training/development splits in Gao et al. (2021). For few-shot learning, the average and standard deviation over 5 different training/development splits are reported. Overall, MAE-LM can be combined with prompt-based methods for effective zero-shot and few-shot learning.

## F   MORE DISCUSSIONS

**Ethical Considerations.** Despite their remarkable performance, pretrained models have been shown to come with risks such as exacerbating harmful biases (Bender et al., 2021; Bommasani et al., 2021). In our experiments, we follow the standard pretraining settings (*e.g.*, data preparation, collection and preprocessing), and we expect more well-documented and filtered text corpora (Dodge et al., 2021), as well as future developments of harm reduction techniques (Liang et al., 2021) may help mitigate the ethical concerns about pretrained models.

**Connections to Prior Work.** Since the advent of BERT (Devlin et al., 2019), there have been numerous developments in new pretraining and fine-tuning methods aiming to improve the effectiveness of pretrained models in downstream tasks. The advantages of these proposed methods, however, are mostly demonstrated via empirical evidence alone, and our understanding of why certain methods are better than the others remains limited. Our analyses in this work may advance the understanding of the benefits of some prominent methods: ELECTRA (Clark et al., 2020) fills [MASK] positions with real tokens; therefore, the encoder does not suffer from the representation deficiency issue. Different from the ablation in Section 4.3 where we randomly sample real tokens to fill [MASK], ELECTRA employs an MLM model to sample replaced tokens which are generally plausible alternatives to the original tokens, thus better preserving the contexts in pretraining. These designs may help partially explain the effectiveness of ELECTRA. Prompt-based methods (Gao et al., 2021; Schick & Schütze, 2021) adapt pretrained MLM models to downstream tasks by creating prompt templates that convert the target task into a masked token prediction problem. This helps mitigate the representation deficiency issue that occurs in standard fine-tuning of MLM models as [MASK] tokens are also introduced into downstream data, resulting in more model dimensions being utilized. Our findings may also shed light on certain previously observed phenomena in MLM models. For example, the rank deficiency issue might be responsible for the de-contextualization in self-attention patterns (Gong et al., 2019).

**Implications on Autoregressive LMs.** While autoregressive LM pretraining generally does not introduce artificial symbols such as [MASK], our analyses can be easily extended to show that the representation deficiency issue can also arise in autoregressive pretraining when certain real tokens exist exclusively in the pretraining data but are either absent or occur infrequently in downstream data. Similar to the impact of [MASK] tokens, these tokens occupy dimensions during pretraining that may not be effectively utilized in downstream tasks. Consequently, it is desirable to maximize the vocabulary overlap between pretraining data and downstream data, which can be realized via pretraining data selection, training corpora pre-processing, and vocabulary pruning. We leave these explorations as future work.

