# OpenReview forum: "Representation Deficiency in Masked Language Modeling"
_ICLR.cc/2024/Conference — ICLR 2024 poster_

### Official Review · Reviewer_HZEv · 2023-10-21

**Soundness:** 2 fair
**Presentation:** 2 fair
**Contribution:** 2 fair
**Rating:** 3
**Confidence:** 4

**Summary:**

The main claim of this paper is that the `[mask]` token takes some dimensions of the representation and that this may raise the risk of overfitting or result in a waste of model capacity. They first provide some empirical results showing that the `[mask]` tokens indeed cause the model to generate lower-rank representations. Theoretically, the authors also show that

- The representation of the `[mask]` has high rank at the last layer. (lemma 2.1)
- The vector space of the real tokens (non-mask tokens) representation at some layer does not include the vector space for the `[mask]` token, so the representation of real tokens can not be full-rank. (theorem 2.2)

These empirical and theoretical analyses motivate them to propose a encoder-decoder-based pretraining approach, MAE-LM, where the encoder’s input does not contain `[mask]` tokens. Their empirical results show that MAE-LM outperform MLM on the GLUE and SQuAD benchmarks.

**Strengths:**

1. They provide empirical evidence supporting their claim.
2. Their proposed MAE-LM outperforms MLM.
3. They conduct comprehensive ablation studies.

**Weaknesses:**

My main concerns are about the theoretical arguments in this paper.

## Main concern 1: The connection between pretraining and fine-tuning

In the introduction, the paper claims that
>  Those dimensions exclusively used for [MASK] tokens have not been pretrained to represent real tokens, and will have to be either trained from scratch on downstream data, raising the risk of overfitting (Hendrycks et al., 2019; Kumar et al., 2022), or become unused, resulting in a waste of model capacity.

I think the interplay between pretraining and fine-tuning is very complicated and not fully understood yet. Thus I don’t think this argument is substantiated. The authors cite Hendrycks et al. (2019) and Kumar et al. (2022) but I am not sure how these two works support this argument. Also, Figure (a) in this paper shows that inputs without a mask token have higher-rank representations. Doesn’t it just indicate that the impact of having mask tokens during pretraining does not impact the rank of real tokens when mask tokens are not used?

## Main concern 2: The unwritten assumptions of Theorem 2.2

***It seems that Theorem 2.2 is based on some unwritten assumptions***, e.g. the model needs to be an attention-only model without MLP and residual layers.

## Main concern 3: The mismatch between the setup for Lemma 2.1 and the setup for Theorem 2.2

Following the previous concern, Theorem 2.2 seems to be  based on some unwritten unrealistic assumptions, while Lemma 2.1 is based on the empirical results that (full) transformers can fit real-world high-rank distributions. If we look at the paper by Dong et al. (2021), we can find that under the assumption Theorem 2.2 is based on, the rank of the representation converges to 1. This implies that, under this assumption, Lemma 2.1 does not hold. Therefore, I think it’s inappropriate to use Theorem 2.2 along with Lemma 2.1 to derive the conclusion of this paper.


## Main concern 4: The weak implication of the theoretical results

The results only suggest that the representation of real tokens cannot be full-rank. It does not characterize how far the representation is from being full-rank. It is possible that the representation of the real tokens has rank $d - 1$. In this case, the so-called “representation deficiency” problem may not be a big issue.


## Minor concern 1: The representation matrix of each example v.s. the whole corpus

The rank of the representation matrices discussed in Lemma 2.1 seems to be the representation matrix of the whole corpus. But in reality, we only feed in the model with a much shorter sequence of tokens, e.g., 512 tokens. In this case, the rank of the distribution of the masked tokens is at most 512 * 15% (in expectation), meaning that the mask tokens can’t occupy too many dimensions.

This may not be a big issue, but I think the arguments starting from Theorem 2.2 need to be rewritten a little bit. For example, in Theorem 2.2, because it’s about the sequence fed into a transformer model, the representation matrix should be of an example rather than of the whole corpus. Therefore, Lemma 2.1 can’t be used directly.


## Minor concern 2: The contribution of this work

This work has some similarities with the work by Dong et al. (2021):

1. Dong et al. (2021) also plot the representation rank of transformer models.
2. Theorem 2.2 in this paper is largely based on the proof from Dong et al. (2021).

I think the authors should give credit to Dong et al. more explicitly.

**Questions:**

I would probably recommend this paper more if the theoretical parts were stated differently (or removed).

---

> ### Author Response · Authors · 2023-11-19
> **Response to Reviewer HZEv**
>
> Thank you very much for your thoughtful feedback! We address your raised points as follows.
>
> **The interplay between pretraining and fine-tuning is very complicated and not fully understood yet**: While we agree that theoretical understanding of the relationship between pretraining and fine-tuning is still lacking, the general consensus in the literature supports the pivotal role of pretraining in enhancing both the effectiveness (Kumar et al.) and robustness (Hendrycks et al.) upon fine-tuning. Therefore, analogous to why a non-pretrained model is inferior to a pretrained counterpart, the model dimensions that are not pretrained to represent real tokens (due to the deficiency issue caused by [MASK] tokens) will be less effective on downstream data. Furthermore, we have shown in Figure 5 (b) that the gap in effective rank persists in fine-tuning, confirming the lasting impact of pretraining.
>
> **Figure (a) in this paper shows that inputs without a mask token have higher-rank representations**: Sorry for the confusion, but our Figure 1 (a) shows the opposite – inputs without [MASK] suffer from lower-rank representations.
>
> **The unwritten assumptions of Theorem 2.2**: In our original submission, we explicitly stated the assumption and rationale for only analyzing the rank change induced by self-attention at the beginning of our proof in Appendix A. In the updated paper version, we have moved them to the main paper for better clarity. In summary, only the rank changes induced by self-attention reflect the extent of contextualization in token representations, and the effectiveness of text encoders is typically attributed to the contextualized representations. While MLPs/residual connections can also change the rank of representations, they do not provide each token with new information from other tokens. While we do not account for MLPs and residual connections, our analysis validates that the rank deficiency is caused by the self-attention mechanism, and in practice, MLPs and residual connections do not prevent the issue from happening.
>
> **The mismatch between the setup for Lemma 2.1 and the setup for Theorem 2.2**: We’d like to clarify that Lemma 2.1 does not impose any architectural assumptions and it holds for any sequence modeling architecture. The empirical studies in Dong et al. (Figure 2 in their paper) are under a different setup than our Lemma 2.1: According to Section 4.1 of Dong et al., the pure attention architecture is not the one directly trained for the MLM task, whereas in our case the model is directly trained with MLM.
>
> **The weak implication of the theoretical results; it is possible that the representation of the real tokens has rank $d-1$**: Our primary goal is to utilize theoretical analysis to substantiate the presence of the representation deficiency issue. We have shown empirically that the representation deficiency issue results in a nontrivial gap in effective rank as large as ~50 dimensions, especially in deeper layers. Furthermore, we demonstrate that addressing this issue via our proposed MAE-LM method yields consistent performance improvements across various downstream tasks. Therefore, the identified issue is a crucial problem in MLM pretraining that warrants attention and remediation.
>
> **The representation matrix of each example v.s. the whole corpus**: Thanks for the question. We’d like to clarify that while the analysis in Lemma 2.1 is conducted at the corpus level, the conclusion of Lemma 2.1 can extend seamlessly to individual sequences. Specifically, if the effective rank of [MASK] token increases throughout Transformer layers when considering the entire corpus, it logically follows that this phenomenon holds true by expectation for the majority, if not all, individual samples in the corpus.
>
> **The contribution of this work and Dong et al.**: While Dong et al. also plot the representation rank, their focus is not on investigating the issue related to [MASK] tokens in MLM pretraining. Regarding our theoretical analysis, we indeed draw inspiration from Dong et al., and we explicitly mentioned this in the full proof in our original submission (Appendix A). In the updated paper, we have added a paragraph at the beginning of our full proof in Appendix A to give Dong et al. more credits.
>
> Thanks again for your thoughtful comments. We have incorporated the revision accordingly in the updated paper. Please let us know if there are any further questions.

---

> > ### Author Response · Authors · 2023-11-22
> > **Looking forward to the discussion**
> >
> > Dear Reviewer HZEv,
> >
> > We sincerely appreciate the time and effort you've devoted to reviewing our work. We understand that your schedule may be quite busy. As the authors-reviewer discussion phase draws to a close, we kindly request your attention to our response. Our aim is to gain insights into whether our response effectively addresses your concerns and to ascertain if there are any additional questions or points you would like to discuss.
> >
> > We look forward to the opportunity for further discussion with you. Thank you for your thoughtful consideration.
> >
> > Best regards, \
> > Authors

---

### Official Review · Reviewer_247c · 2023-10-30

**Soundness:** 4 excellent
**Presentation:** 4 excellent
**Contribution:** 3 good
**Rating:** 10
**Confidence:** 5

**Summary:**

This paper identifies a significant discrepancy in Masked Language Modeling (MLM) pretraining, where model representations are skewed towards the [MASK] token, absent in downstream tasks. To address this, the authors introduce MAE-LM, a novel algorithm that enhances model efficiency and performance, outstripping robust baselines across various metrics.

**Strengths:**

1. The paper effectively pinpoints an important yet overlooked issue in BERT-like language models' (LMs) pretraining, specifically the representation mismatch due to [MASK] tokens. It then proceeds to look into this mismatch, attributing it to rank deficiency with robust empirical and theoretical backing in Section 2.2. This perspective is both insightful and novel.

2. The proposed method, MAE-LM, addresses the identified mismatch issue without resorting to complex architectural alterations. This simplicity ensures the resulting pretrained model retains compatibility with existing BERT-like models, demonstrating the method's applicability.

3. The paper's evaluation is very solid, adhering to standard practices with comprehensive benchmarking on GLUE and SQuAD, compared against multiple strong baselines. MAE-LM consistently outperforms these baselines across multiple scales—base, base++, and large++—sometimes even by substantial margins.

4. Detailed analyses and studies confirm that the performance boost really stems from improved model dimension utilization, which is a direct result of addressing the rank deficiency issue.

5. The writing and presentation of the paper are top-notch, mirroring the high quality of its content.

**Weaknesses:**

1. Although the paper mentions in Footnote 2 that some MLM training settings (like Google's original BERT) keep 10% of [MASK] tokens as original, and that subsequent studies like Wettig et al. (2022) found this trick unnecessary, an exploration of the effective rank of models using this 80:10:10 technique, similar to Figure 1, would be beneficial.

2. Missing reference: [this paper](https://proceedings.mlr.press/v97/gong19a.html) looked into attention patterns in pretrained BERT-like LMs, uncovering patterns related to low-rankness. This study could provide additional insights into the topic at hand.

**Questions:**

Could you explore the effective rank of models using 80:10:10 masking technique (like Google's original BERT)?

---

> ### Author Response · Authors · 2023-11-19
> **Response to Reviewer 247c**
>
> Thank you very much for your thoughtful feedback, and we appreciate your positive acknowledgment of our work! We address your raised points as follows.
>
> **Using 80:10:10 masking technique**: In our preliminary experiments, we found that the 80:10:10 masking technique did not provide notable performance gain compared to directly masking out 15% of the tokens, and thus we used the latter in our evaluations. That said, we agree that validating our findings under the 80:10:10 masking technique is beneficial. We have added the effective rank plot in Appendix E, Figure 6 of our updated paper. The results are largely similar to Figure 1, which confirms that randomly replacing a small percentage of [MASK] tokens with real ones does not effectively address the representation deficiency issue, as the ratio of [MASK] tokens in pretraining is still high.
>
> **Missing reference**: Thank you for pointing out the reference, and we have added the discussions of this work to both Section 5 and Appendix F. We agree that our findings in this paper are related to the paper you mentioned: The rank deficiency problem might have caused de-contextualization in token representations, which is reflected by the localized self-attention patterns as discovered in that study.
>
> Thanks again for your review! Please let us know if you have any further questions.

---

### Official Review · Reviewer_sv4z · 2023-11-06

**Soundness:** 4 excellent
**Presentation:** 4 excellent
**Contribution:** 4 excellent
**Rating:** 8
**Confidence:** 4

**Summary:**

This paper studies the following discrepancy: for masked language models (MLMs), the [MASK] token is only presented in pre-training but not in downstream fine-tuning. Based on both empirical and theoretical analysis, the authors show that the presence of [MASK] token in pretraining occupy some model dimensions even in downstream applications when [MASK] is not used, that is, not all model dimensions are leveraged to represent tokens other than [MASK]. To address this discrepancy, the authors propose MAE-LM, removing [MASK] tokens from MLM pre-training. Empirical evaluations show consistent improvement on the GLUE and SQuAD benchmarks compared to well known MLMs including BERT and RoBERTa.

**Strengths:**

Being able to identify this discrepancy and provide a clear and in-depth analysis is the major strength of this paper. The proposed solution to this problem is simple and effective, providing insights for more sophisticated approaches.

This paper is overall very well presented; in particular, the theoretical analysis presented in this paper are easy to follow and immediately to-the-point; Lemma 2.1 shows that the rank of the matrices for [MASK] token representation increases as the layers go deeper, providing a nice explanation to the empirical observation; Theorem 2.2 shows that the embedding of some [MASK] tokens need to be orthogonal to that of real tokens, limiting the number of dimensions that can be used.

**Weaknesses:**

It would be nice if the authors can expand a little bit more on the implications of this work on autoregressive LMs but I really do not see any major weakness in this paper. Though someone may argue that MAE-LM’s improvements on benchmarks are relatively small compared to baselines, this is not a quite problem as the improvements shown are quite consistent; besides, one major contribution of this paper is really to identify the discrepancy between pre-training and downstream fine-tuning and to analyze its effect on MLM’s expressiveness.

**Questions:**

n/a

---

> ### Author Response · Authors · 2023-11-19
> **Response to Reviewer sv4z**
>
> Thank you very much for your thoughtful feedback, and we appreciate your positive acknowledgment of our work! We discuss the implications of our work on autoregressive LMs as follows.
>
> While autoregressive LM pretraining generally does not introduce artificial symbols such as [MASK], our analyses can be extended to show that the representation deficiency issue can also arise in autoregressive pretraining when certain real tokens exist exclusively in the pretraining data but are either absent or occur infrequently in downstream data. Similar to the impact of [MASK] tokens, these tokens occupy dimensions during pretraining that may not be effectively utilized in downstream tasks. Consequently, it is desirable to maximize the vocabulary overlap between pretraining data and downstream data, which can be realized via pretraining data selection, training corpora pre-processing, and vocabulary pruning. We leave these explorations as future work, and we have added the discussions to Appendix F in the updated paper.
>
> Thanks again for your review! Please let us know if you have any further questions.

---

> > ### Comment · Reviewer_sv4z · 2023-11-22
> >
> > Thank you for providing your perspective on autoregressive models. My score remains unchanged.

---

### Official Review · Reviewer_3FYT · 2023-11-08

**Soundness:** 4 excellent
**Presentation:** 4 excellent
**Contribution:** 3 good
**Rating:** 6
**Confidence:** 4

**Summary:**

This paper employs effective rank to analyze the representation deficiency caused by the [mask] token in Masked Language Models (MLM). Based on the analysis results, this paper proposes the MAE-LM model, which does not input [mask] in the encoder and supplements [mask]  in the shallow decoder. The MAE-LM model achieves satisfactory results in downstream tasks.

**Strengths:**

1. This work is the first study using effective rank to analyze the decrease in expressive power caused by the [mask] token, leading to two insightful theorems. These theorems may inspire future research in the field of MLM.

2. This paper provides rigorous mathematical proofs supporting the analytical findings.

3. The selected tasks for experimentation are representative, and the promising experimental results demonstrate a performance improvement of MAE-LM over classical MLM models.

**Weaknesses:**

This paper follows the classic framework of analysis + improvement, but both core modules closely resemble existing work.

1. In the analysis module, the use of effective rank to represent MLM's representative capacity overlaps with section 3.1 of ISOTROPY (https://openreview.net/pdf?id=xYGNO86OWDH), where effective rank is used to denote the isotropy of representations. The differences between these two approaches are minor.

2. In the experimental module, the model structure used in this paper is identical to the one used in Mask Later (https://arxiv.org/pdf/2211.04898.pdf).

**Questions:**

The authors should explicitly define how this work differs from existing works, otherwise,  it appears as if the paper merely uses the theoretical tool of ISOTROPY to analyze why Mask Later is effective. Without a distinct contribution, the overall impact of this work seems limited.

---

> ### Author Response · Authors · 2023-11-19
> **Response to Reviewer 3FYT**
>
> Thank you very much for your thoughtful feedback! We address your raised points as follows.
>
> **The use of effective rank to represent MLM's representative capacity overlaps with section 3.1 of ISOTROPY**: We’d like to clarify that we follow the definition of effective rank in ISOTROPY (Cai et al.) to only compute the rank and empirically showcase the identified representation deficiency issue, and we have not used it in our theoretical analyses. Our major contributions are (1) investigating the discrepancy between pretraining and fine-tuning of MLM models caused by [MASK] tokens from the perspective of representation rank, and (2) demonstrating that a simple architectural modification in MLM—excluding [MASK] tokens from the encoder’s input—can effectively address the representation deficiency and result in improved downstream performance. These contributions are orthogonal to Cai et al. We have updated our paper to provide clarifications in Section 2.2.
>
> **The model structure used in this paper is identical to the one used in Mask Later**: Compared to the 3ML approach proposed by Liao et al., our proposed MAE-LM shares the same masked autoencoder architecture. However, the fundamental motivations, focuses, results and insights of the two works are different:
> * _Focuses_: Liao et al. focus on an efficiency aspect, by excluding [MASK] tokens to reduce the sequence length. In contrast, our primary goal is to use principled empirical and theoretical analyses to illuminate an overlooked issue in MLM—specifically, the impact of [MASK] tokens on the model's representation power. The MAE-LM framework is introduced as a simple solution to rectify the rank deficiency issue arising from [MASK] tokens, aiming for the understanding of its influence on the learned representations and the downstream task performance.
> * _Results_: In contrast to Liao et al.'s efficiency-driven emphasis, our experimental results substantiate that MAE-LM effectively mitigates the representation gap induced by [MASK] tokens, consistently outperforming standard MLM across various tasks and pretraining settings. To our knowledge, such a representation learning benefit of the masked autoencoder architecture has not been explored before.
> * _Insights_: Importantly, our contributions may extend beyond the specific model architecture and offer insights for future developments in MLM pretraining. For example, one may regularize the [MASK] token representations to reside in the same space of real token representations as another potential solution to the representation deficiency problem. Our analyses also underscore the importance of considering real tokens that exist solely in pretraining data, yet are absent or occur rarely in downstream data. Such tokens, akin to [MASK] tokens, have the potential to compromise model representation power.
>
> We have incorporated a brief discussion in Section 3 of our revised paper to emphasize the distinctiveness of our contribution.
>
> Thanks again for your review! Please let us know if you have any further questions.

---

> ### Comment · Reviewer_3FYT · 2023-11-20
> **Re: Response to Reviewer 3FYT**
>
> Thank you for your prompt response.
>
> 1. **Analysis moudle**. After carefully reviewing your reply and Appendix A, I have reconsidered the theoretical analysis section of the paper. I recognize that both lemmas in the paper are non-trivial, and the proof methods used differ from ISOTROPY. So my concerns about the analysis module have been well addressed.
>
> 2. **Experiment moudle.** Nevertheless, my concerns persist regarding the experiment module. Despite different motivations, Liao et al.'s results have already demonstrated that MAE-LM can achieve performance comparable to, or even superior to, traditional MLM (as shown in Table 3 of Liao et al.'s paper). In other words, we already hold a prior expectation that, regardless of the correctness of the theoretical analysis in this paper, the MAE-LM structure will work.
>
> However, I fully agree that the theoretical analysis of this paper "may extend beyond the specific model architecture and offer insights for future developments in MLM pretraining." Consequently, I have raised the score to 6.

---

> > ### Author Response · Authors · 2023-11-22
> > **Thank you for the reply**
> >
> > Thank you for the prompt response. We would like to express our sincere gratitude for your constructive feedback. Your opinions have significantly elevated the quality and clarity of our paper. Regarding the experiment module, we will make further revision accordingly to make it more clear. Thank you for the comments!
> >
> > Best, \
> > Authors

---

### Author Response · Authors · 2023-11-19
**General Response**

We sincerely thank all the reviewers for their thoughtful comments and feedback. We are grateful for the recognition of our work in various aspects:

* The problem studied in this work is important and novel (247c), and the theoretical analyses are insightful, rigorous (3FYT, 247c), easy to follow, and immediately to-the-point (sv4z)
* The proposed method is simple and effective (3FYT, sv4z, 247c, HZEv)
* The empirical evaluation is comprehensive (HZEv) and sound (247c, 3FYT)
* The presentation is clear (sv4z, 247c)

Based on the reviewers' suggestions, we have updated our paper with key changes highlighted in blue. We summarize these revisions as follows:

* Add clarifications of our contributions to Section 2.2 and Section 3 (3FYT)
* Add discussions of the implications of our work on autoregressive LMs to Appendix F (sv4z)
* Add the results under the BERT masking strategy to Appendix E and add discussions of a related study to Section 5 and Appendix F (247c)
* Move the assumption of Theorem 2.2 from Appendix A to the main paper (HZEv)

---

### Meta-Review · Area_Chair_J2SN · 2023-12-06

**Metareview:**

The paper under review, "Improving Masked Language Models Using Effective Rank Analysis," proposes a novel approach to addressing the representation deficiency caused by the [MASK] token in Masked Language Models through the introduction of the MAE-LM model. The paper has garnered attention for its theoretical and empirical strengths and has been reviewed thoroughly by reviewers. The writing and presentation of the paper are commended for their clarity and conciseness.

Strengths:

1. The work is the first to use effective rank to diagnose and analyze the drop in expressive power due to the [MASK] token in MLMs. It presents two insightful theorems which the reviewers agree may shape future research in the MLM field.

2. It includes rigorous mathematical proofs bolstering the findings, thus appreciably advancing the theoretical understanding of MLMs.

3. The empirical evaluation is robust and the proposed MAE-LM showcases consistent performance gains over classical MLM models, which is demonstrated extensively using widely recognized benchmarks like GLUE and SQuAD.


Weakness:

1. The reviewers noted that the paper largely builds upon the conceptual frameworks of ISOTROPY and Mask Later. Yet, they acknowledge that the authors have integrated their insights in a novel way. After an exchange, it is clear that while there is an overlap, the paper presents a distinct and significant contribution, and the author should make these clear in the revised paper.

2. Despite different motivations, Liao et al.'s results have already demonstrated that MAE-LM can achieve performance comparable to, or even superior to, traditional MLM.

3. Exploring implications for autoregressive LMs could be valuable.

4.  As mentioned by Reviewer HZEv, the whole analysis is valid only for an attention-only transformer model and it is necessary to highlight this and give a detailed discussion to avoid misleading in the theoretical part.

**Justification For Why Not Higher Score:**

Although there are some differences, many of the ideas in this paper have already been introduced in existing work.  This paper explicitly summarizes and distills these contents, which is also valuable.

**Justification For Why Not Lower Score:**

The paper fulfills the criteria for clear presentation, sound methodology, and a meaningful contribution to the literature, and clocking robust improvements in widely recognized benchmarks also illustrates its practical relevance.  3 of 4 reviewers voted for acceptance. Reviewer HZEv pointed out some issues in the theoretical part, which should be addressed in the revised manuscript.

---

### Decision · Program_Chairs · 2024-01-16

Accept (poster)